# Staufen2-mediated RNA recognition and localization requires combinatorial action of multiple domains

Simone Heber[1,2], Imre Gáspár [3,6], Jan-Niklas Tants[4], Johannes Günther[4], Sandra M. Fernandez Moya[5], Robert Janowski[2], Anne Ephrussi [3], Michael Sattler [2,4] & Dierk Niessing [1,2]

Throughout metazoans, Staufen (Stau) proteins are core factors of mRNA localization particles. They consist of three to four double-stranded RNA binding domains (dsRBDs) and a C-terminal dsRBD-like domain. Mouse Staufen2 (mStau2)-like *Drosophila* Stau (dmStau) contains four dsRBDs. Existing data suggest that only dsRBDs 3–4 are necessary and sufficient for mRNA binding. Here, we show that dsRBDs 1 and 2 of mStau2 bind RNA with similar affinities and kinetics as dsRBDs 3 and 4. While RNA binding by these tandem domains is transient, all four dsRBDs recognize their target RNAs with high stability. Rescue experiments in *Drosophila* oocytes demonstrate that mStau2 partially rescues dmStau-dependent mRNA localization. In contrast, a rescue with mStau2 bearing RNA-binding mutations in dsRBD1–2 fails, confirming the physiological relevance of our findings. In summary, our data show that the dsRBDs 1–2 play essential roles in the mRNA recognition and function of Stau-family proteins of different species.

[1] Institute of Pharmaceutical Biotechnology, 89081 Ulm University, Ulm, Germany. [2] Institute of Structural Biology, Helmholtz Zentrum München, 85764 Neuherberg, Germany. [3] Developmental Biology Unit, European Molecular Biology Laboratory, 69117 Heidelberg, Germany. [4] Center for Integrated Protein Science Munich at Chair of Biomolecular NMR Spectroscopy, Department Chemistry, Technische Universität München, 85747 Garching, Germany. [5] Biomedical Center Munich, Department of Cell Biology, Ludwig-Maximilians-Universität München, 82152 Planegg-Martinsried, Germany. [6] Present address: Institute of Molecular Biotechnology, 1030 Vienna, Austria. Correspondence and requests for materials should be addressed to D.N. (email: dierk.niessing@uni-ulm.de)

mRNA localization is an essential mechanism for a range of cellular processes, including embryonic development, cell differentiation, and migration, as well as neuronal plasticity[1]. For active transport of mRNAs along the cellular cytoskeleton, ribonucleoprotein particles (RNPs) are formed. Such mRNA-containing RNPs (mRNPs) consist of motor proteins, RNA binding proteins, helicases, and translational regulators[2].

In the mature nervous system, mRNA localization to pre- and postsynaptic areas followed by local translation has been implicated in memory and learning[3,4]. For instance, dendritically localized RNAs produce proteins with synaptic functions such as $Ca^{2+}$/calmodulin kinase II (CaMKII), the cytoskeletal protein Arc or microtubule-associated protein 2 (MAP2), and AMPA or NMDA receptors.

The RNA-binding protein Staufen (Stau) was originally identified in *Drosophila* as an mRNA transport factor required to establish the anterior–posterior axis of the embryo[5,6]. Together with proteins of the exon-junction complex (EJC) and the translational repressor Bruno it binds to *oskar* mRNA, which is transported from the nurse cells to the oocyte and then localized to its posterior pole[7]. During *Drosophila* neurogenesis, the asymmetric segregation of *prospero* mRNA into the ganglion mother cell requires Stau function as well[8].

In mice, the two Staufen-paralogs mStau1 and 2 share about 50% protein-sequence identity and have both been implicated in mRNA localization and RNA-dependent control of gene expression[9–11]. Whereas, mStau1 is ubiquitously expressed and required for Staufen-mediated decay (SMD) of its target mRNAs via UPF1 interaction, mStau2 expression is enriched in the heart and brain[12–16]. The two mammalian paralogs Stau1 and Stau2 were reported to bind distinct, yet overlapping sets of target mRNAs[10,17], indicating distinct but possibly complementary functions. Consistent with this hypothesis is the observation that although both paralogs appear to mediate degradation of RNAs, only Stau2 seems to also stabilize a subset of its target mRNAs[18].

A transcriptome-wide analysis of *Drosophila* Stau (dmStau) targets suggested certain RNA-secondary structure elements as Stau-recognized structures (SRSs)[19]. A subsequent study in mice used immunoprecipitation- and microarray-based experiments to identify the *Regulator of G-Protein Signaling 4* (*Rgs4*) mRNA as an mStau2-regulated transcript and found two predicted SRS stem-loops in its 3′UTR[18].

All Stau-family proteins contain multiple so-called double-stranded RNA-binding domains (dsRBD). Whereas mStau1 contains three dsRBDs, a tubulin-binding domain (TBD) and one C-terminal noncanonical dsRBD-like domain, mStau2 and dmStau have four dsRBDs, followed by a tubulin-binding domain (TBD) and a C-terminal, noncanonical dsRBD-like domain[20].

For all mammalian Stau proteins, the dsRBD3 and dsRBD4 are thought to be required and sufficient for full target mRNA binding[11,12,21], whereas dsRBDs 1, 2, and 5 are often referred to as pseudo dsRBDs, which retained the fold but not activity of canonical dsRBDs[21]. The longest isoform of Stau2, Stau2[62], is most similar to dmStau, both possessing all five dsRBDs. Stau2[62] shuttles between the nucleus and cytoplasm and has been proposed to transport RNAs from the nucleus to distal dendrites[22]. Because Stau dsRBDs only seem to interact with the backbone of RNA[23] and do not undergo recognizable sequence-specific interactions[24], one of the unresolved questions is how specific RNA binding can be achieved by dsRBD 3–4.

Here, we show that in mStau2, the noncanonical dsRBDs 1 and 2 exhibit RNA-binding activity of equal affinity and kinetic properties as the known RNA-binding dsRBDs 3–4. Mutational analyses and biophysical characterization of RNA binding revealed that dsRBD 1–2 have to act in concert with dsRBD 3–4

to allow for stable, high-affinity RNA binding. Using *Drosophila* as model system, we demonstrate the importance of RNA binding by dsRBDs 1–2 for Stau function in vivo and show that mStau2 can partially substitute for dmStau function during early *Drosophila* development. The requirement of a combination of two dsRBD-tandem domains and thus the possibility of binding to two stem loops allows recognition of combinations of secondary structure and thus a much more complex readout for specific binding. This observation might help to explain how Stau proteins can bind selectively to their RNA targets in vivo.

## Results

**mStau2 binds directly to SRS motifs in the *Rgs4* 3′UTR.** To probe a potential direct interaction between mStau2 and the *Rgs4* mRNA, we performed in vitro binding experiments with mStau2 and the two previously predicted SRS motifs of the *Rgs4* 3′UTR (Supplementary Fig. 1a, Supplementary Table 1). EMSAs with full-length mStau2 showed binding with apparent equilibrium dissociation constants ($K_D$) in the low micromolar concentration range for *Rgs4* SRS1 as well as for SRS2 (Fig. 1a). The entire 3′UTR of *Rgs4* mRNA was bound by full-length mStau2 with higher affinity (Supplementary Fig. 1b). Deletion of SRS1 and SRS2 did not reduce RNA binding, indicating that regions other than the SRS motifs contribute to mStau2 binding. Since EMSAs with very long RNAs do not yield very precise results, we also performed experiments with a 3′UTR fragment consisting of 634 bases of the *Rgs4* 3′UTR (Rgs4-mini) that contains both predicted SRSs. EMSAs with the previously reported RNA binding dsRBDs 3–4 using either wild-type *Rgs4*-mini RNA or a mutant version, in which SRS1 and SRS2 were deleted, showed similar affinities (Supplementary Fig. 1c).

These observations indicated that other cryptic SRSs might be present in the *Rgs4* 3′UTR. In the *Rgs4*-mini RNA, another region was predicted to fold into a stable imperfect stem loop with 26 paired bases interrupted by two bulges. Although this stem loop is longer and predicted to be more stable, it could still serve as cryptic SRS. Surprisingly, mStau2 bound to this stem loop with a $K_D$ in the 100 nM range (Fig. 1a). This stem loop is termed SRS* (Supplementary Fig. 1a, Supplementary Table 1).

**The mStau2 tandem domain dsRBD 1–2 binds dsRNA.** Next, we tested whether binding is indeed only mediated by dsRBD 3–4, as indicated by previous studies[12,21]. Surprisingly, in EMSAs mStau2 dsRBD 1–2 bound the SRS2 RNA with an affinity comparable with that of dsRBD 3–4 (Fig. 1b). This finding shows that dsRBDs 1 and 2 are not inactive pseudo dsRBDs as previously suggested, but contribute to RNA recognition of mStau2. Upon increased protein concentrations, a supershift was observed, indicating either binding of additional dsRBDs to the RNA or an oligomerization of the protein itself. Binding of mStau2 dsRBD 1–2 to SRS2 RNA was further confirmed by NMR titration experiments (Supplementary Fig. 2). Upon addition of the stem-loop RNA to the tandem domain dsRBD 1–2, chemical shift changes and differential line broadening of NMR signals in the protein and the RNA are observed. To investigate the binding interface of mStau2 dsRBD 1–2 on SRS2 RNA, imino signals of the unbound RNA were compared with the respective resonances when bound to mStau2 at a equimolar ratio. Significant line broadening was observed in the imino signals of the four base pairs close to the stem terminus (U4, U27, U29, and G30). These were most strongly affected in the NMR spectrum of the complex (Fig. 1c), whereas other imino signals were less affected. The differential line broadening indicates binding kinetics in the intermediate exchange regime on the chemical shift time scale[25,26].

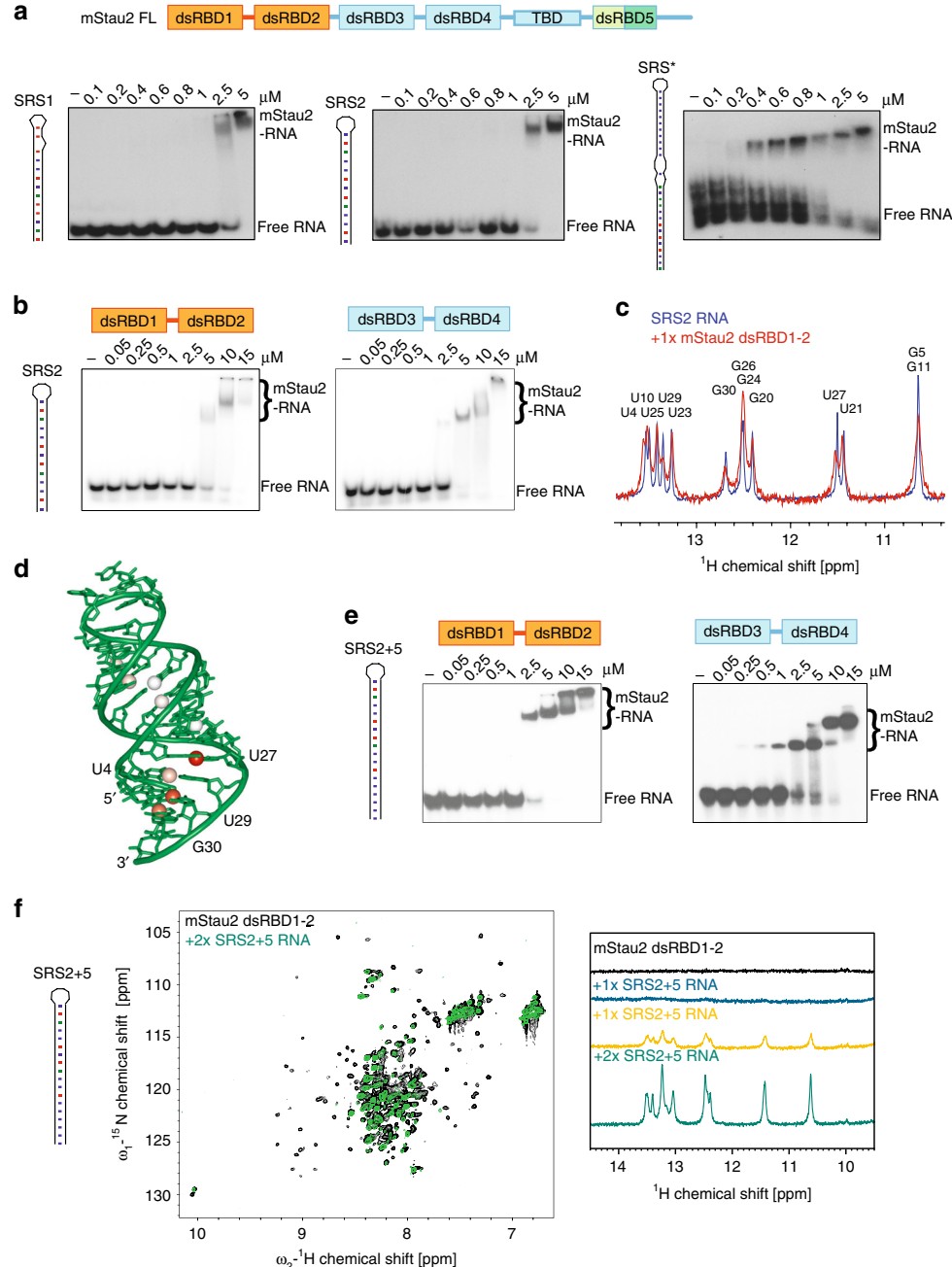

**Fig. 1** The dsRBD1–2 tandem domain of mStau2 binds RNA. **a** EMSAs with full-length mStau2 (mStau2 FL) and different SRS RNAs from the 3′UTR of the *Rgs4* mRNA. **b** EMSAs with mStau2 tandem domains dsRBD1–2 or dsRBD3–4 and SRS2. **c** NMR titration experiments of mStau2 dsRBD1–2 with SRS2 RNA. **d** Crystal structure of SRS2 at 1.73 Å resolution. Iminos from **c** showing significant line broadening are indicated by colored (orange to red) spheres. **e** EMSAs with mStau2 tandem domains dsRBD 1–2 or dsRBD 3–4 and SRS2 RNA extended by five basepairs (SRS2 + 5). **f** NMR titration experiments of mStau2 dsRBD 1–2 with SRS2 + 5 RNA. Left: overlay of $^{1}$H,$^{15}$N-HSQC spectra of dsRBD1–2 in absence and presence of 2x excess SRS2 + 5 RNA. Resonance shifts and line broadening of several signals are observed. Right: comparison of 1D imino traces of SRS2 + 5 RNA at different stoichiometric ratios with dsRBD 1–2. Strong line broadening of imino signals is observed in presence of protein. Source data are provided as a Source Data file

To unambiguously confirm the dsRNA fold of *Rgs4* SRS2, we solved its crystal structure at 1.73 Å resolution (Fig. 1d, Table 1). The RNA adopts a typical A-form double-stranded helix, characterized by a wide and shallow minor groove and a deep and narrow major groove. Whereas electron density in the stem region of both molecules is very well defined, the density map in the loop region is poor (Supplementary Fig. 3), indicating flexibility of the RNA in this region. Imino signals observed in imino $^{1}$H,$^{1}$H-NOESY spectra are consistent with the base pairing observed in the crystal structure (Supplementary Fig. 4). Our

NMR data and the crystal structure thus confirm that SRS2 folds into a canonical stem-loop structure.

**Role of the length of dsRNA for mStau2 tandem domain binding.** To test the effect of stem-loop length on RNA binding, the SRS2 stem was extended by five basepairs (SRS2 + 5, see Supplementary Fig. 1a). EMSAs with dsRBD 1–2 or with dsRBD 3–4 showed significantly improved binding (Fig. 1e), indicating that the length of the stem has great influence on the affinity. In NMR titration experiments with dsRBD 1–2, amino acids affected

upon SRS2 + 5 RNA binding seem identical to SRS2 binding (Fig. 1f; compare with Supplementary Fig. 2). 1D imino traces of the RNA, however, revealed line broadening at substoichiometric concentrations for SRS2 + 5 dsRBD 1–2 binding (Fig. 1f compare with Fig. 1c). Strong line-broadening of all imino signals suggests dynamic binding involving sliding of the dsRBDs on the RNA helix, as previously shown for other dsRBDs[27–29]. With the shorter SRS2 stem-loop RNA, all imino signals are observable at equimolar RNA:protein ratio. Line broadening for the imino signals in the basepairs at the bottom of the stem suggests this as a main interaction region. Because the protein may not slide off the hairpin end but rather gets stopped there, protein binding to a hairpin RNA is expected to introduce some asymmetry to binding

and thus differential line broadening. In contrast, the 18 bp stem of SRS2 + 5 allows for significant sliding as reflected by the severe line broadening observed for all imino signals in the basepairs of the stem upon protein binding.

To determine whether the loop region of the RNA is required for stem-loop recognition by mStau2 tandem domains, we tested the SRS2 stem elongated by five base pairs, but lacking its loop (SRS2 + 5Δloop). This elongated stem was bound by both tandem domains dsRBD 3–4 and dsRBD 1–2 with affinities similar to the original SRS2 stem-loop (Supplementary Fig. 5a; compare with Fig. 1b, e), indicating that the loop region is not essential for RNA recognition. After showing that the affinity of mStau2 to RNA correlates with the length of dsRNA, we aimed to define the minimal length of the RNA stem required for recognition by mStau2 tandem domains. Both tandem domains dsRBD 1–2 and dsRBD 3–4 bound to RNA stem loops comprising stems of 12 bp, 10 bp, and 8 bp with similar affinities, with apparent dissociation constants ($K_{D}s$) in the micromolar concentration range (Supplementary Fig. 5b–d). Only when the stem was decreased to 7 bp, binding was almost abolished (Supplementary Fig. 5e). Thus, a stem of 8 bp appears to be the minimal length required for recognition by mStau2 tandem domains. Of note, available structures of dsRBDs show binding to longer RNA stem loop of about 19 bp length[30,31]. It is therefore well possible that our observed binding to a minimal stem-loop RNA only reflects a partial recognition and that for a full binding a longer stem is required. This interpretation is consistent with our general observation that longer RNAs are bound stronger than shorter ones.

**Kinetics of mStau2 RNA binding.** In order to understand the kinetics of mStau2 binding, surface plasmon resonance (SPR) experiments with biotin-labeled SRS* RNA or SRS2 + 5 RNA coupled to a streptavidin sensor chip surface were performed. For the tandem domains dsRBD 1–2 and dsRBD 3–4, rapid binding and dissociation kinetics were observed for both RNAs, already at the lowest tested concentration of 10 nM (Fig. 2a, b). Because of the fast kinetics, the on- and off-rates could not be accurately quantified. However, the steady-state binding is best described by a two-site binding fit for dsRBD 1–2 with a $K_{D}1$ of 130 nM for the SRS2 + 5 RNA and of 25 nM for the SRS* RNA (Fig. 2a; Table 2). $K_{D}2$ could not be determined because binding was not saturated at the highest measured concentration of 1 μM. Because mStau2 tends to oligomerize at low micro-molecular concentrations, even higher concentration ranges could not be tested.

---

**Table 1 Data collection and refinement statistics (molecular replacement)**

|  | SRS2 RNA |
|---|---|
| Data collection |  |
| Space group | C 1 2 1 |
| *Cell dimensions* |  |
| a, b, c (Å) | 114.020, 32.390, 46.370 |
| α, β, γ (°) | 90.00, 103.47, 90.00 |
| Resolution (Å) | 55.44 - 1.73 |
| I/σ(I) | 17.92 (1.88) |
| CC$_{1/2}$ | 99.9 (75.7) |
| Completeness (%) | 96.8 |
| (in resolution range) | (39.82 - 1.73) |
| Redundancy | 5.1 (5.4) |
| *Refinement* |  |
| Resolution (Å) | 39.82 - 1.73 |
| No. of reflections | 16,965 |
| R$_{work}$/R$_{free}$ | 18.5/23.7 |
| R$_{free}$ test set | 819 reflections (4.83 %) |
| *No. of atoms* | 1421 |
| RNA | 1290 |
| Ba ion | 13 |
| Mg ion | 1 |
| Water | 116 |
| Wilson *B* factor (Å²) | 30.6 |
| Average B, all atoms (Å²) | 45.0 |
| Anisotropy | 0.048 |
| Fo,Fc correlation | 0.97 |
| *R.m.s. deviations* |  |
| Bond lengths (Å) | 0.007 |
| Bond angles (°) | 0.910 |

---

**Table 2 mStau2 binding to SRS* and SRS2 + 5 RNA**

| mStau2 wt | Binding | Kinetics | $K_{D}1$ [nM] | $K_{D}2$ [nM] | Hill coefficient |
|---|---|---|---|---|---|
| *SRS* RNA* |  |  |  |  |  |
| FL | Two-site | Stable | 10.6 ± 5 | 195 ± 103 |  |
| dsRBD3–4 | Two-site | Transient | 9 ± 1 | n.d. (>1000) | – |
| dsRBD1–2 | Two-site | Transient | 25 ± 8 | n.d. (>1000) | – |
| dsRBD1–4 | Hill | Stable | – | 287 ± 127 | 1.7 ± 0.4 |
| dsRBD2 | Hill | Transient | – | 828 ± 29 | 1.1 ± 0.04 |
| dsRBD1 | No binding | – | – | – | – |
| *SRS2 + 5* |  |  |  |  |  |
| FL | Two-site | Stable | 10.6 ± 5 | 195 ± 103 |  |
| dsRBD3–4 | Two-site | Transient | 18 ± 16 | n.d. (>1000) | – |
| dsRBD1–2 | Two-site | Transient | 130 ± 30 | n.d. (>1000) | – |
| dsRBD1–4 | Hill | Stable | – | 330 ± 148 | 1.6 ± 0.2 |
| dsRBD2 | Hill | Transient | – | 650 ± 247 | 1.3 ± 0.3 |
| dsRBD1 | no binding | No binding | – | – | – |

± indicates standard deviation

---

Also, dsRBD 3–4 bound with similar properties, yielding a $K_D 1$ of 18 nM for SRS2 + 5 RNA and of 9 nM for the SRS* RNA (Fig. 2b; Table 2). As with dsRBD 1–2, $K_D 2$ was in the micromolar range and could not be determined. Together these findings confirm the RNA-binding activities of dsRBD 1–2 and of dsRBD 3–4, with similar binding properties. The observed fast kinetics for $K_D 1$ explain why in EMSAs, no high-affinity band shifts were observed.

Interestingly, when the SPR experiments were repeated with mStau2 dsRBD 1–4, the binding kinetics changed dramatically with both RNAs. Binding as well as dissociation occurred at much slower rates, indicating that the formed complexes are stable (Fig. 2c; Table 2). Steady-state affinities could no longer be described by a two-site binding fit, most likely due to higher-order binding events by mStau2′s four dsRBDs. However, when using Hill-fit $K_D$s of 357 nM and 330 nM for the SRS2 + 5 RNA and the SRS* RNA were obtained, respectively. Together with observed Hill coefficients of $n \geq 1.7$, these data indicate cooperative binding, which results in the formation of stable mStau2–RNA complexes. Whether this cooperativity arises from interactions of individual dsRBDs within one protein or from protein–protein interaction between different molecules cannot be determined from these data. The previously reported dimerization of Stau1[32] suggests that mStau2 might also form oligomers. We did, however, not detect oligomerization of full-length mStau2 by SEC-SLS (Supplementary Fig. 6) and thus consider cooperativity by intermolecular interactions unlikely.

Finally, we assessed binding of the SRS2 + 5 and SRS* RNAs to full-length mStau2. Consistent with our RNA-binding experiments with dsRBD1–4 (Fig. 2c), in both cases stable complexes were formed. Steady-state binding was described by two-site binding fits with nanomolar affinities of $K_D 1 = 1.3$ nM and $K_D 2 = 185$ nM for SRS2 + 5 and of $K_D 1 = 10.6$ nM and $K_D 2 = 195$ nM for the SRS* RNA (Fig. 2d; Table 2).

**The individual dsRBDs 1 and 2 bind RNA dynamically**. In order to obtain structural insights into RNA-binding preferences of dsRBDs 1 and 2, $^1$H,$^{15}$N-HSQC NMR spectra of the individual dsRBDs and of the tandem dsRBD 1–2 were measured. The spectra of the two individual domains show that they are well-folded (Supplementary Fig. 7) and also nicely match with the NMR spectrum of the tandem domain dsRBD 1–2. This indicates that in the context of the tandem domains, the structures of the individual dsRBDs 1 and 2 are not altered and do not significantly interact with each other.

Upon titration of SRS2 + 5 RNA to the individual dsRBDs 1 and 2, chemical shift perturbations and line broadening are observed (Fig. 3). Residues affected by RNA binding to the isolated dsRBDs are similar to those seen in titration experiments with the dsRBD 1–2 tandem domain (Fig. 3a, c; compare with Fig. 1f), suggesting that both domains bind the RNA independently.

One-dimensional imino spectra of the RNA upon protein binding indicate line broadening at substoichiometric concentrations for dsRBD2 (Fig. 3d) similar to what was observed for the tandem domain dsRBD 1–2 (Fig. 1f). Interestingly, for dsRBD1, less line broadening is observed for imino signals (Fig. 3b). This indicates that the two dsRBDs bind RNA with different kinetics, which is suggestive of a lower binding affinity of dsRBD1 compared with dsRBD2.

**dsRBD1 binds RNA significantly weaker than dsRBD2**. In order to understand the respective contribution of each dsRBD, we performed SPR experiments with the individual dsRBDs. Surprisingly, at protein concentrations up to 1 µM no RNA

binding was observed for dsRBD1 (Fig. 4a, b; Table 2). In contrast, dsRBD2 bound to SRS2 + 5 RNA and to SRS* RNA with $K_D$s of 650 nM and of 829 nM, respectively (Fig. 4c, d; Table 2). This binding was observed with fast on- and off-rates, similar to the tandem domain (Fig. 2a, b). Furthermore, dsRBD2 shows no sign of cooperativity, as indicated by Hill coefficients close to 1 (Fig. 4c, d). The lack of detectable RNA binding by SPR experiments with dsRBD1 compared with the detected interaction in NMR titrations can be explained by the much higher RNA concentrations used in the NMR experiments (50 µM).

We confirmed these findings by EMSAs, in which dsRBD1 did not bind to SRS2 + 5 RNA (Supplementary Fig. 8a). Also, dsRBD2 bound SRS2 + 5 RNA much weaker than the tandem domain dsRBD1–2, as binding was observed only at concentrations >10 µM (Supplementary Fig. 8a).

The two domains are connected by a linker region of 19 amino acids. Thus, an explanation for the stronger binding of the tandem domain could be that their linker region contributes to the RNA binding of one dsRBD. We tested this possibility by performing EMSAs either with a dsRBD1-linker fragment or with a fragment consisting of linker-dsRBD2. In neither of these cases did we observe any improved binding to SRS2 + 5 RNA (Supplementary Fig. 8b). Also, mixing the two individual dsRBDs with linker did not improve RNA-binding activity (Supplementary Fig. 8c). Together these results indicate that the two domains act in concert to bind dsRNA with better affinities and that this requires the presence of the linker, which itself does not appear to contribute to the RNA recognition.

**Mutations in dsRBD1 moderately impair dsRBD 1–2 RNA binding**. For further verification of the observed binding properties and to allow for functional in vivo studies of the RNA-binding activity of mStau2 dsRBD 1–2, RNA-binding mutants of the dsRBDs 1 and 2 were designed. For dsRBD1, mutations were introduced based on the NMR titration experiments and multiple sequence alignments with dmStau (Supplementary Fig. 9a). A partial assignment allowed for the identification of residues with chemical shift perturbations upon RNA titration, pointing at their location within or close to the binding interface. These residues map to the predicted end of helix $\alpha_1$, loop 2, and the beginning of helix $\alpha_2$, which are the regions that mediate RNA binding in a canonical dsRBD (Supplementary Fig. 9a). Conserved dsRBD residues close or within these regions were chosen for mutation. We mutated glutamate in helix $\alpha_1$ (E15), histidine in loop 2 (H36), lysines from the conserved KKxxK motif (K59 and K60), and phenylalanine in the beta strand $\beta_2$ (F40). Mutation of these residues in dsRBD3 from *D. melanogaster* to alanines had been shown to abolish RNA binding completely[23].

The dsRBD1–2 tandem domain with a range of mutations in dsRBD1 were tested for binding to SRS2 + 5 (Supplementary Fig. 10; Table 3). For the mutations E15A, H36A, F40A, K59A, K60A, and K59A K60A binding kinetics were fast. Except for E15A, the steady-state binding curves are best described by Hill-fits with Hill coefficients $n \approx 1$, indicating non-cooperative binding. Whereas the observed $K_D$s of dsRBD 1–2 H36A and K59A are similar to that of dsRBD2 alone, dsRBD 1–2 mutations F40A, K60A, and K59A K60A bind with even lower affinity than dsRBD2 alone. These results indicate that binding activity of dsRBD1 was abolished by these mutations. The only exception was dsRBD 1–2 E15A, where steady-state binding to SRS2 + 5 was fitted with a first-order binding reaction and a $K_D$ of 132 nM, indicating that RNA-binding activity of dsRBD1 might be compromised but not completely abolished.

When the same dsRBD1 mutations in the context of the dsRBD 1–2 fragment were tested for binding to the SRS* RNA,

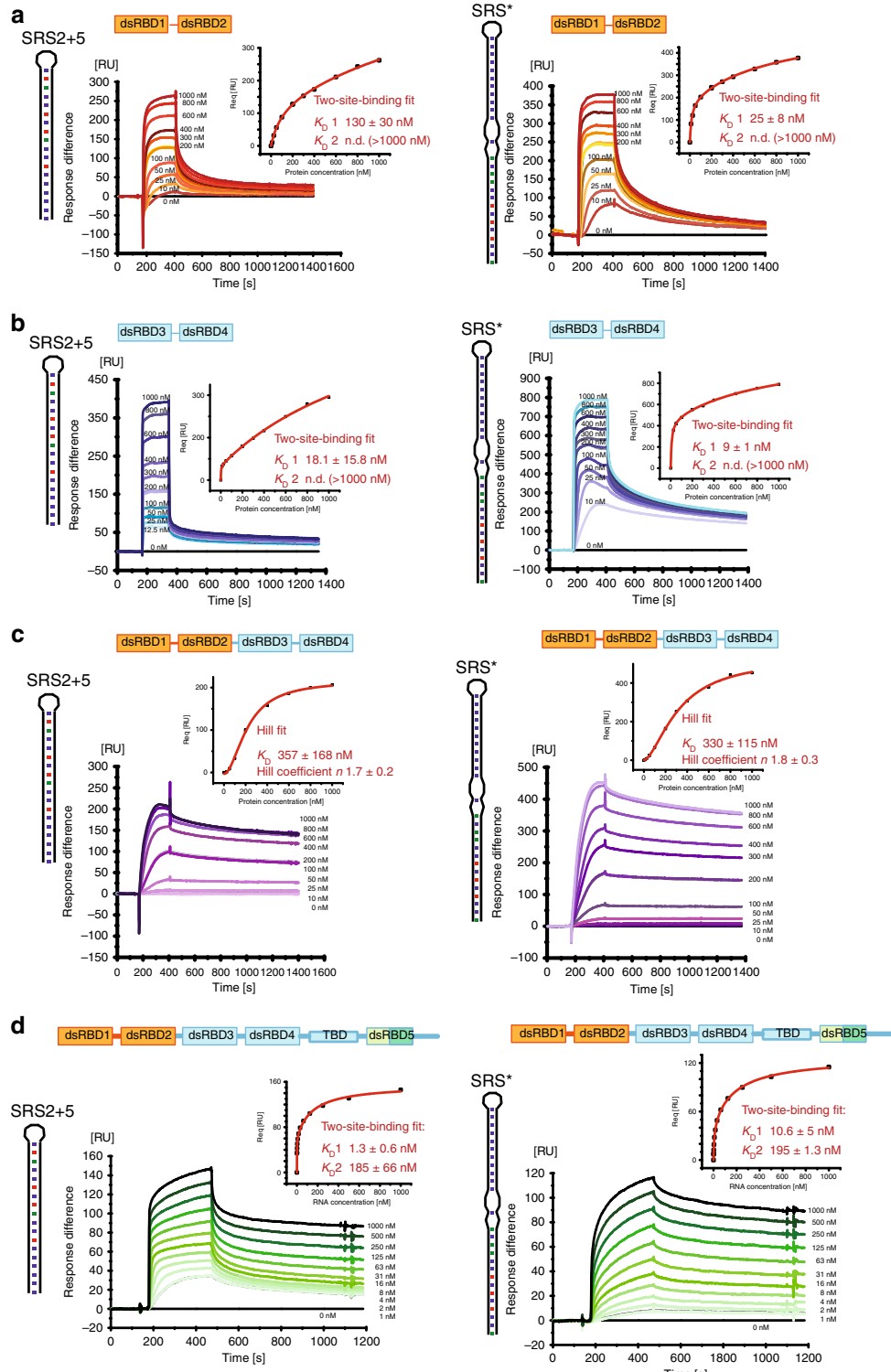

**Fig. 2** SPR shows mStau2 binding to SRS2 + 5 RNA and SRS* RNA. **a** mStau2 dsRBD 1–2, and **b** mStau2 dsRBD 3–4 binding to surface-coupled SRS2 + 5 and SRS* RNAs. The tandem domains dsRBD 1–2 and dsRBD 3–4 bind transiently with fast kinetics. The steady-state binding curves do not saturate up to 1 μM protein concentration but can be described by a two-site binding fit with $K_D1$ of 18 nM and 130 nM, respectively, for SRS2 + 5 and $K_D1$ of 9 nM and 25 nM, respectively, for SRS*. **c** mStau2 dsRBD 1–4 binding to surface-coupled SRS2 + 5 and SRS* RNAs is stable with slower kinetics. The steady-state binding curve saturates at approximately 1 μM and is described by a Hill fit with an apparent overall $K_D$ of 357 nM and a Hill coefficient $n = 1.7$ for SRS2 + 5 and an apparent overall $K_D$ of 330 nM and a Hill coefficient $n = 1.8$ for SRS*, indicating positive cooperative binding. **d** SRS2 + 5 and SRS* bind to surface-coupled mStau2 FL stably and with high affinity. The steady-state binding curves can be described by a two-site binding fit with $K_D1$ of 1.3 nM and 10.6 nM and $K_D2$ of 185 nM and 195 nM, respectively. ± indicates standard deviation. Source data are provided as a Source Data file

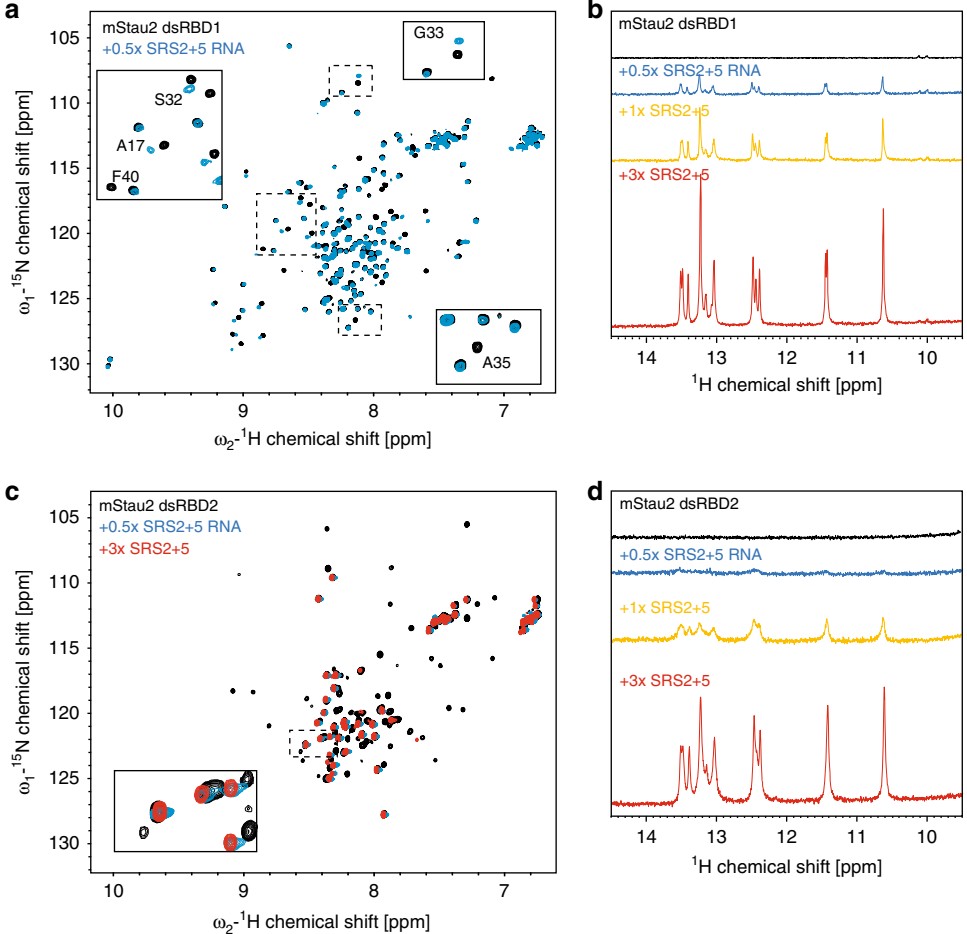

**Fig. 3** NMR titrations of mStau2 dsRBD1 and dsRBD2 with SRS2 + 5 RNA. **a**, **c** Overlay of $^1$H,$^{15}$N-HSQC spectra of dsRBD1 (**a**) or dsRBD2 (**c**) in absence and presence of SRS2 + 5 RNA. Resonance shifts and line broadening of several signals are observed for both domains. Note, that there are two sets of signals observed for dsRBD1, where only one set is affected by RNA binding. The second set of signals may reflect the presence of an alternate conformation of a region of dsRBD1. **b**, **d** Comparison of 1D imino NMR spectra of SRS2 + 5 RNA at different stoichiometric ratios with dsRBD1 (**b**) or dsRBD2 (**d**). Strong line broadening of imino signals is observed in presence of dsRBD2 but not dsRBD1, pointing at reduced RNA binding affinity for dsRBD1

the mutations H36A, F40A, K59A, and K59A K60A behaved again very similarly to their binding to SRS2 + 5, confirming that RNA-binding activity of dsRBD1 is abolished by these mutations (Supplementary Fig. 11; Table 3). However, dsRBD 1–2 E15A binds to SRS* RNA similar to the wild-type protein, showing two-site binding with $K_D1 = 15$ nM and $K_D2 = 405$ nM (Supplementary Fig. 11a; compare with Fig. 2a), indicating that RNA-binding activity of dsRBD1 is not corrupted by this mutation. Also, dsRBD 1–2 K60A bound SRS* RNA, unlike SRS2 + 5 RNA, with affinities similar to the wild-type protein (Supplementary Fig. 11d). This mutation possibly has a less drastic effect.

**Mutations in dsRBD2 impair RNA binding of dsRBD 1–2.** Due to the lack of NMR assignments for dsRBD2, to design mutations in this domain, we had to rely on sequence homology. A sequence alignment of 12 species was used to identify conserved, positively charged, or aromatic residues for mutation (Supplementary Fig. 9b). These residues, E99A, K106A, F157A, and H169A, were individually mutated in the context of the dsRBD 1–2 tandem domain, and subsequently tested for RNA binding by SPR. All mutants showed strongly decreased binding to SRS2 + 5 (Supplementary Fig. 12; Table 3) and fitting of binding curves indicated that two-site binding was lost in all mutants. All dsRBD2

mutations in the context of dsRBD 1–2 were also tested for binding to SRS* RNA. Unlike SRS2 + 5, mStau2 dsRBD 1–2 E99A and K106A bound SRS* with properties similar to the wild-type protein (Supplementary Fig. 13, Table 3), indicating that the effects of these mutations are less dramatic. Binding of dsRBD 1–2 H169A, however, was still strongly decreased, such that a $K_D$ could not be determined. Binding of dsRBD 1–2 F157A was again strongly impaired and no $K_D$ could be determined, thus confirming the results obtained for SRS2 + 5.

**Mutations in dsRBD 1 and 2 impair RNA binding of dsRBD 1–2.** Based on the SPR results for the single-point mutations in dsRBD1 and dsRBD2, double-mutants were designed in the context of mStau2 dsRBD 1–2. In dsRBD1, the mutation F40A was chosen because it had a strong effect on binding to both tested RNAs, its resonance shifted upon RNA titration in the $^1$H,$^{15}$N-HSQC spectra, and it is conserved in dmStau. In dsRBD2, the mutations F157A and H169A were chosen. F157A showed altered binding kinetics, and H169A had the strongest effect on RNA binding of all tested dsRBD2 mutations. Both residues are conserved in dmStau. Correct folding of double-mutant proteins was verified by CD spectroscopy (Supplementary Fig. 14). As expected, in the SPR experiments all double-mutant versions of

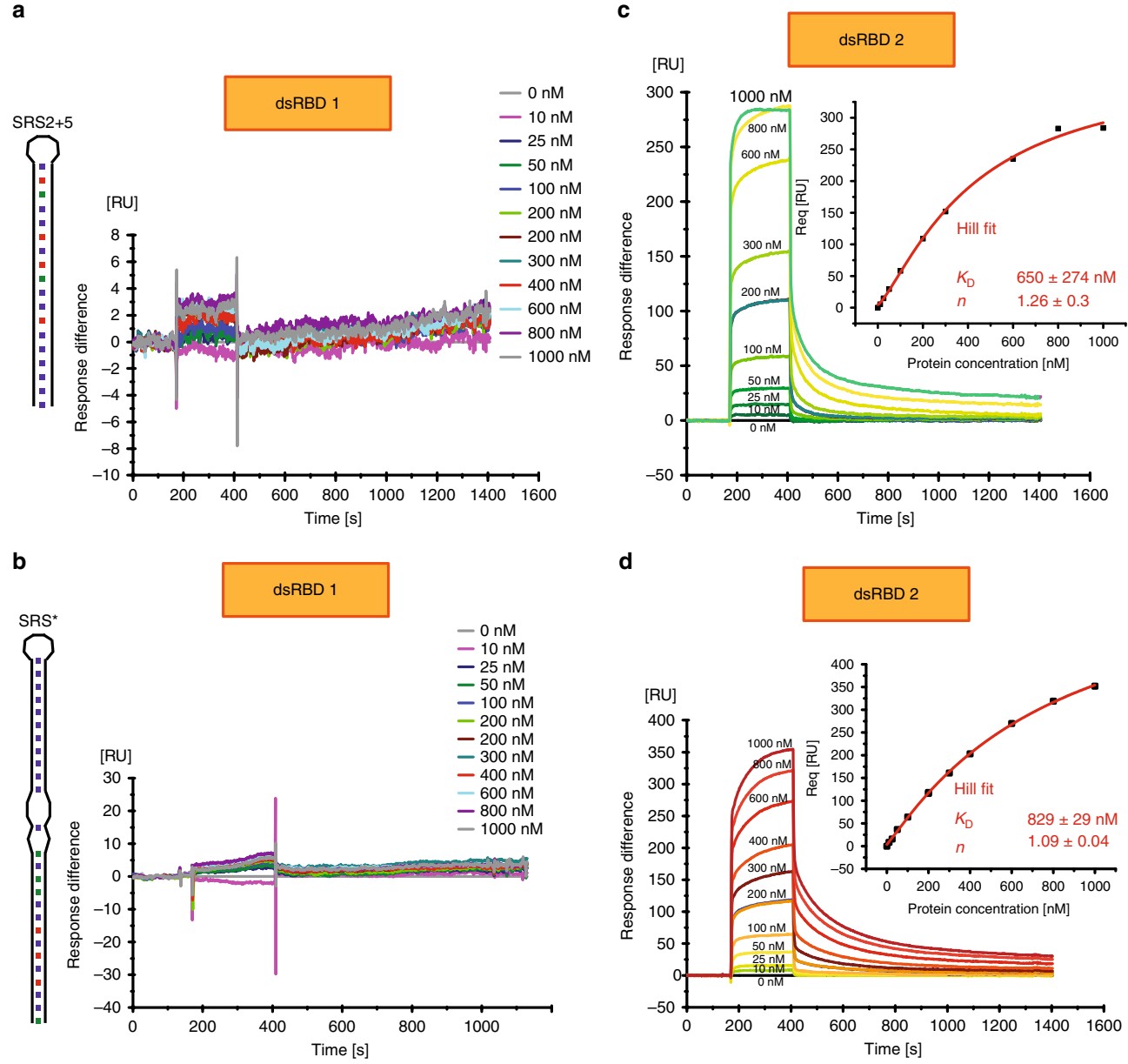

**Fig. 4** SPR experiments with mStau2 dsRBD1 or dsRBD2. At low micromolar concentrations, mStau2 dsRBD1 alone binds neither to (**a**) SRS2 + 5 nor to (**b**) SRS*. In contrast, dsRBD2 binds to (**c**) SRS2 + 5 and to (**d**) SRS* with fast kinetics in a non-cooperative fashion with $K_D$s of 650 nM or 829 nM, respectively. Steady-state binding curves are described by a Hill fit. ± indicates standard deviation. Source data are provided as a Source Data file

mStau2 dsRBD 1–2 lacked binding to SRS2 + 5 and to SRS* RNAs (Supplementary Fig. 15; Table 3).

**Mutations in dsRBD1 and 2 impair RNA binding of dsRBD 1–4.** To assess the contribution of dsRBD 1–2 to RNA binding in the context of all four verified RNA-binding dsRBDs, SPR experiments were performed with mStau2 dsRBD 1–4. Comparison of the wild-type mStau2 dsRBD 1–4 with a dsRBD 1–4 fragment harboring the double mutation F40A F157A (Fig. 5a) showed that RNA binding by the mutant protein was significantly impaired (Fig. 5b, d; compare with Fig. 2c).

In contrast to wild-type dsRBD 1–4, the mStau2 dsRBD 1–4 F40A H169A bound SRS2 + 5, and SRS* RNA with fast binding kinetics, resembling the transient binding by dsRBD 3–4 alone (Fig. 5c, e; Table 3; compare with Fig. 2b). The steady-state binding is well described by a two-site binding fit with $K_D$s resembling those of dsRBD 3–4 alone. In addition, kinetic fits to

the binding curves obtained at 1000 nM protein were performed. A bivalent analyte fit to the binding curve at 1000 nM shows that the rate-constants $ka_1$ and $kd_1$ are both significantly increased for F40A F169A when compared with the wild-type protein (Supplementary Fig. 16). Taken together, this indicates that in dsRBD 1–4 F40A H169A RNA binding is mediated by dsRBD 3–4 alone. Since mutations in dsRBD 1–2 impair the affinity of dsRBD 1–4 and because the interactions become much more transient, our data indicate that for efficient and stable RNA binding of mStau2 all four dsRBDs have to act in concert.

**Rescue experiments with mStau2 in *Drosophila* Stau$^{−/−}$ embryos.** To assess the relevance of our findings for the in vivo function of Stau proteins, we took advantage of the well-studied role of Stau in early *Drosophila* development. dmStau is required for intracellular transport of *oskar* mRNA within the oocyte for its localization and translational activation at the posterior pole of

**Table 3 mStau2 dsRBD1–2 mutant binding to SRS\* and SRS2 + 5 RNA**

| mStau2 dsRBD1–2 protein | Binding | Kinetics | $K_D1$ [nM] | $K_D2$ [nM] | Hill coefficient |
|---|---|---|---|---|---|
| *S*RS* RNA | | | | | |
| wt | Two-site | Transient | 25 ± 8 | n.d. (>1000) | – |
| E15A | Two-site | Transient | 15 ± 11 | 406 ± 359 | – |
| H36A | Hill | Transient | – | 499 ± 149 | 0.9 ± 0.06 |
| F40A | Hill | Transient | – | 769 ± 427 | 1 ± 0.1 |
| K59A | Hill | Transient | – | 624 ± 142 | 0.9 ± 0.2 |
| K60A | Two-site | Transient | 36 ± 5 | n.d. (>1000) | – |
| K59A K60A | Hill | Transient | – | n.d. (>1000) | 1 ± 0.1 |
| E99A | Two-site | Transient | 14 ± 0.7 | 850 ± 295 | – |
| K106A | Two-site | Transient | 30 ± 10 | n.d. (>1000) | – |
| F157A | Hill | Stable | – | n.d. (>1000) | 1.2 ± 0.2 |
| H169A | Hill | Transient | – | n.d. (>1000) | 1 ± 0.6 |
| F40A F157A | No fit | Transient | – | – | – |
| F40A H169A | No fit | Transient | – | – | – |
| *SRS2 + 5 RNA* | | | | | |
| wt | Two-site | Transient | 130 ± 30 | n.d. (>1000) | – |
| E15A | Hill | Transient | – | 132 ± 31 | 1 ± 0.1 |
| H36A | Hill | Transient | – | 701 ± 386 | 1 ± 0.2 |
| F40A | Hill | Transient | – | n.d. (>1000) | 1.2 ± 0.1 |
| K59A | Hill | Transient | – | 509 ± 136 | 1.2 ± 0.2 |
| K60A | Hill | Transient | – | n.d. (>1000) | 1 ± 0.1 |
| K59A K60A | Hill | Transient | – | n.d. (>1000) | 1 ± 0.02 |
| E99A | Hill | Transient | – | 635 ± 308 | 1.5 ± 0.3 |
| K106A | No fit | Transient | – | n.d. (>1000) | – |
| F157A | Hill | Stable | – | 497 ± 238 | 1.5 ± 0.2 |
| H169A | No fit | Transient | – | n.d. (>1000) | – |
| F40A F157A | No fit | Stable | – | – | – |
| F40A H169A | No fit | Transient | – | – | – |
| ± indicates standard deviation | | | | | |

the oocyte[7]. We expressed GFP-dmStau, GFP-mStau2, GFP-mStau2[F40A H169A], and GFP-mStau2[F40A F157A] transgenes in the germline of *stau^R9/stau^D3* mutant females[5] that lack endogenous dmStau protein and characterized their ability to rescue the mutant phenotype. As expected, wild-type GFP-dmStau supported efficient accumulation of *oskar* mRNA at the posterior pole (Fig. 6a–d) and even hatching of the majority of the resulting larvae. Also, mStau2-expressing *stau^R9/stau^D3* mutant flies showed localization of a moderate fraction of *oskar* mRNA to the posterior pole of stage 9 oocytes (Fig. 6a–d). At stage 10, the localization of *oskar* mRNA improved further. However, no hatching larvae were observed. mStau2 bearing the double mutation [F40A H169A] or [F40A F157A] in its dsRBDs 1 and 2 was equally expressed (Supplementary Fig. 17), but failed to localize *oskar* mRNA to the posterior pole beyond *stau* null levels (Fig. 6a–d). Evaluation of the content of mRNPs revealed a clear correlation between dmStau protein and *oskar* mRNA copy number (Fig. 6e, f). To a lesser extent, GFP-mStau2 also showed this positive correlation (Fig. 6e, f), suggesting an interaction between the mammalian Stau protein and *oskar* mRNA. In contrast, the mutant proteins largely failed to scale with *oskar* mRNA copy number (Fig. 6f). In the case of *bicoid* mRNPs, mStau2 copy number per mRNA scaled like wild type in the case of [F40A H169A], but failed to scale in the case of [F40A F157A] (Fig. 6g, h).

In summary, these rescue experiments confirm the importance of RNA binding by dsRBD 1–2 for the in vivo function of Stau proteins.

## Discussion
Previous reports had identified dsRBDs 3 and 4 in Stau proteins as the RNA-binding domains[11,12,32], suggesting that dsRBDs 1 and 2 fulfill other functions. In our present work, we have demonstrated that the mStau2 dsRBDs 1 and 2 also possess RNA-binding activity. Furthermore, we could show that the two dsRBDs 1 and 2 work together as a tandem domain to achieve their full functionality. Our data further confirm that also dsRBD 3–4 act as a tandem domain. Our comparison of RNA-binding affinities of single domains and tandem domains bearing mutations suggests that the first binding event with moderate affinity is achieved by the second dsRBD in each tandem domain, namely dsRBD2 and dsRBD4.

Based on our results, we propose a model in which sequential binding events lead to stable RNA recognition by Stau (Fig. 7). In this model, binding of the first tandem domain occurs initially at a random position, with dsRBD2 achieving the first interaction (Fig. 7a, left side). Subsequently, dsRBD1 also binds, thereby increasing the affinity of the tandem domain to dsRNA (Fig. 7a, right side). For a longer RNA stem, the tandem domains bind in a dynamic fashion to the RNA helical stem as indicated by line broadening observed in the NMR experiments. In the tandem domain dsRBD 3–4, the second domain, dsRBD4, binds with higher affinity (Supplementary Fig. 18) and thus likely undergoes the first priming contact in a fashion similar to dsRBD2 (Fig. 7b, left side). Then, the other, free dsRBD of the tandem domain also joins the RNA-bound complex (Fig. 7b, right side). Only when the two tandem domains dsRBD 1–2 and dsRBD 3–4 act together does the protein form a stable complex with RNA. This interpretation is consistent with the stronger and more stable RNA binding of dsRBD 1–4 and full-length mStau2. While we cannot exclude specificity of dsRBDs for certain sequence motifs, we found no experimental evidence for such an assumption. Our data rather suggest that the mStau2 protein recognizes its RNA target in a structure- and not a sequence-dependent manner.

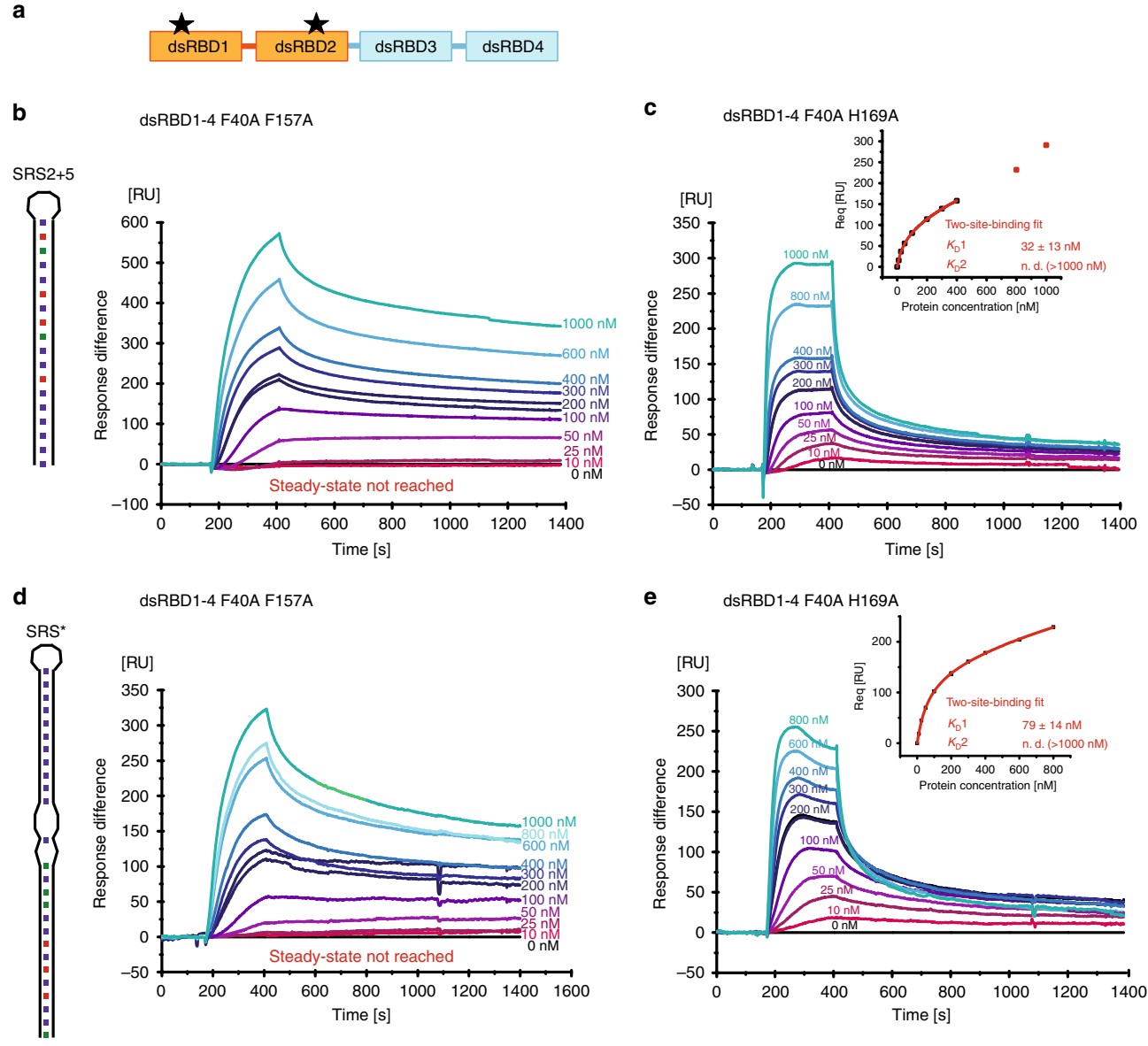

**Fig. 5** SPR with mutated dsRBD 1–4 confirms a contribution of dsRBD1–2. **a** Schematic drawing of mStau2 with its two mutations in dsRBD 1–2. **b**, **c** SPR experiments with mStau2 dsRBD 1–4 double-mutants binding to SRS2 + 5 RNA and **d**, **e** to SRS*. Binding to (**b**) SRS2 + 5 and (**d**) SRS* is strongly decreased for the F40A F157A mutant as compared to mStau2 dsRBD 1–4 wild-type. Binding of the F40A H169A mutant shows dramatically altered kinetics (**c**, **e**) as compared with dsRBD 1–4 wild-type and resembles binding by the tandem domain dsRBD 3–4 alone. ± indicates standard deviation. Source data are provided as a Source Data file

Likely, scenarios for the specificity reported in vivo include the recognition of combinations of secondary structure elements or a contribution of cofactors.

To confirm in vivo that RNA binding by the dsRBDs 1–2 is important for the function of the full-length protein, we utilized the *Drosophila* oocyte as model system. In the germline of otherwise *stau* null flies, different variants of Stau were expressed and the rescue of the mutant phenotype assessed by analyzing *oskar* mRNA localization to the posterior pole. Surprisingly, in mStau2-expressing oocytes, a moderate rescue of *oskar* mRNA localization was observed. In contrast, mStau2-rescue constructs bearing RNA-binding mutations in dsRBD 1–2 failed to rescue *oskar* localization. This observation confirms the importance of dsRBD 1–2 for RNA binding and RNA localization in vivo.

This observation, together with the fact that the long isoform of mStau2 has the same number of dsRBDs, indicate that mStau2 might be the functional homolog of dmStau. The observed mild

rescue, however, indicates differences between the two proteins regarding their specificities for target RNAs or cofactors. Also, the dsRBD 1–5 of the house fly (Musca domestica) Stau failed to rescue all aspects of *oskar* mRNA localization[33]. It will be interesting to see the basis of these functional differences in future experiments.

The fact that mStau1 lacks the first dsRBD raises the question how this paralog achieves full binding. One option is that its mode of RNA binding is different enough from mStau2 to allow for strong and stable binding even with only two or three dsRBDs. An alternative could be that the reported dimerization of mStau1[32] allows for the joint action of dsRBDs in trans and thus full, stable RNA binding is achieved.

Sliding as an initial binding mode also occurs in other RNA binding proteins such as the *Drosophila* protein Loqs-PD, a member of the siRNA silencing pathway. Loqs-PD contains two canonical dsRBDs that show highly dynamic binding and

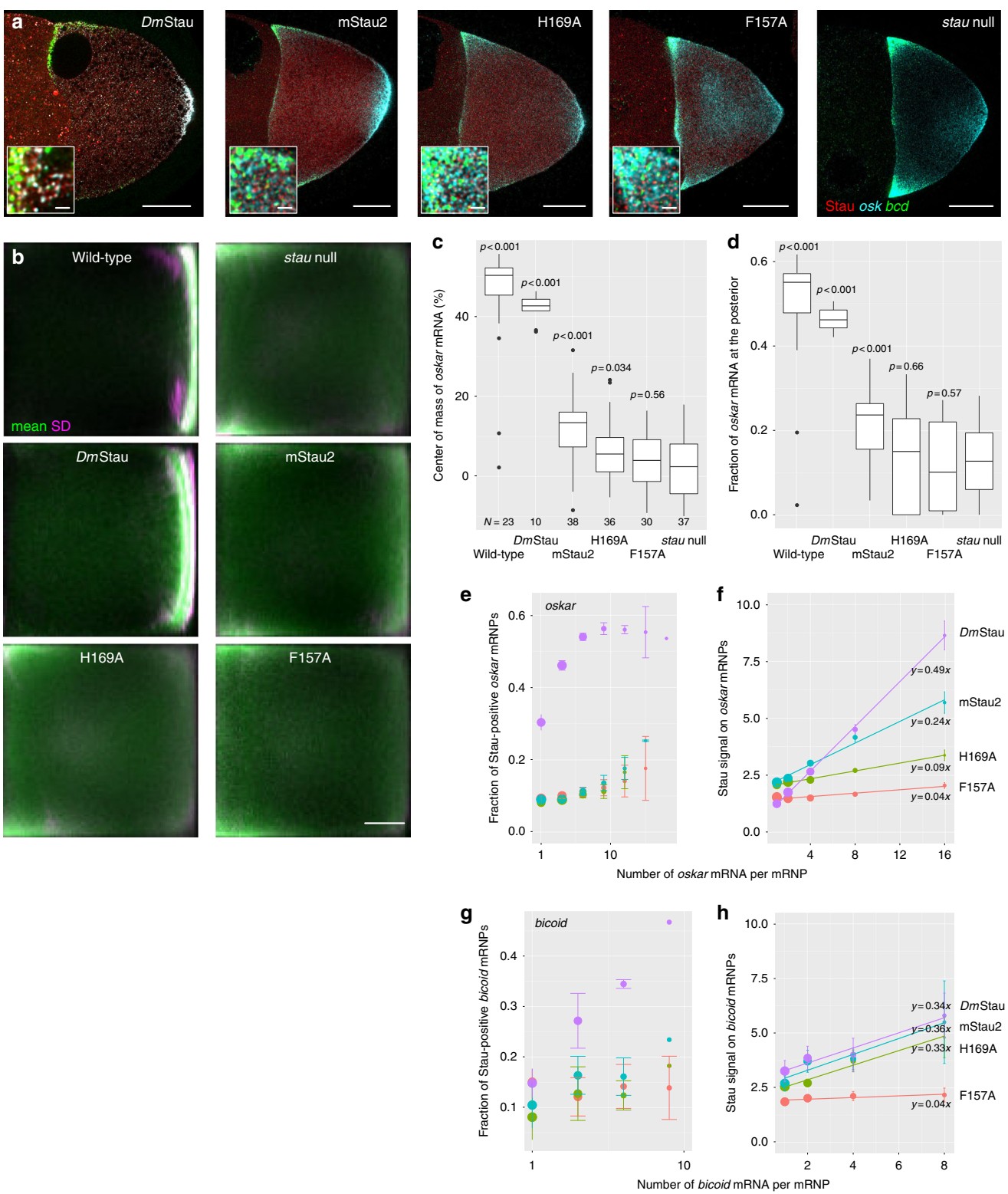

involves sliding along RNA stems[28]. A similar activity was also reported for the human ortholog of Loqs-PD, TAR RNA-binding protein (TRBP)[27]. In contrast to these examples, however, mStau2 involves two tandem domains with four dsRBDs for its sequential RNA-target recognition.

The feature of stable RNA binding is likely to be of great importance for transport of transcripts over longer distances. It is therefore not surprising that our rescue experiments of *Drosophila stau* mutants with Stau constructs required all four dsRBDs

to be functional. The presented model (Fig. 7) offers a mechanistic view on how mStau2 may recognize biological targets with high affinity and stability. Future work will have to answer whether a defined spatial arrangement of two stem-loops is recognized by each tandem dsRBD or if all four domains act as a molecular ruler for a single stem loop of defined length.

RNA-binding proteins in higher eukaryotes very often contain multiple RNA-binding domains[34]. It is thought that these act in a combinatorial fashion such as we have shown for the dsRBDs of

**Fig. 6** Functional interaction of mStau2 with *oskar* mRNA in *Drosophila*. **a** Expression of GFP-dmStau, GFP-mStau2, GFP-mStau2[F40A H169A] and GFP-mStau2[F40A F157A] in germline of *stau^R9^/stau^D3^* mutant females. In dmStau-expressing oocytes *oskar* (cyan) localizes almost exclusively to the posterior pole (right) and *bicoid* (green) to the anterior pole (left) during stage 9 of oogenesis. Transgenic GFP-Stau protein is shown in red. In oocytes lacking Stau (*stau* null), *oskar* is found at both poles, enriching slightly more at the anterior, while *bicoid* localization is unaffected. Insets show magnified regions of the upper anterior corner. Scale bar: 20 μm and 1 μm for insets. **b** Typical localization of *oskar* mRNA in oocytes as function of expressed Stau protein. Using image transformation algorithms, RNA signal was redistributed into a 100 × 100 square matrix and statistically evaluated to obtain average (green) and variability (magenta) of RNA distribution. In wild-type oocytes (top left) most signal is found close to the posterior pole (right of the panels) by stage 9. In absence of Stau (stau null), *oskar* mRNA accumulates at the anterior pole. Scale bar: 20% length of anteroposterior axis. **c, d** Center of mass (relative to geometric center at 0, c) and fraction at posterior pole of *oskar* mRNA (**d**) during stage 9. *P*-values show result of pairwise Mann–Whitney U tests vs. the *stau* null condition (Bonferroni corrected alpha value: 0.01). *N* = number of oocytes. Center line: median; box limits: 25^th^ and 75^th^ percentile; whiskers: 10^th^ and 90^th^ percentile. **e–h** Interaction of GFP-tagged Stau molecules with *oskar* (**e, f**) and *bicoid* (**g, h**) mRNPs. mRNPs are sorted by their mRNA content using quantitative smFISH. Fraction of Stau positive mRNPs (**e, g**) and normalized GFP-Stau signal intensity (**f, h**) were plotted as function of mRNA content of the mRNPs. The normalized GFP-Stau signal intensities are fitted linear models, with indicated slopes. In pairwise comparisons of *oskar* mRNPs (**f**), all slopes are significantly different (p < 0.0001), except for GFP-mStau2[F40A H169A] vs GFP-mStau2[F40A F157A] (*p* = 0.016, *alpha_corrected* = 0.01). In *bicoid* mRNPs (**h**), the slope of GFP-mStau2[F40A F157A] differs from the other three (*p* < 0.01), which have similar slopes (*p* > 0.9). Source data are provided as a Source Data file

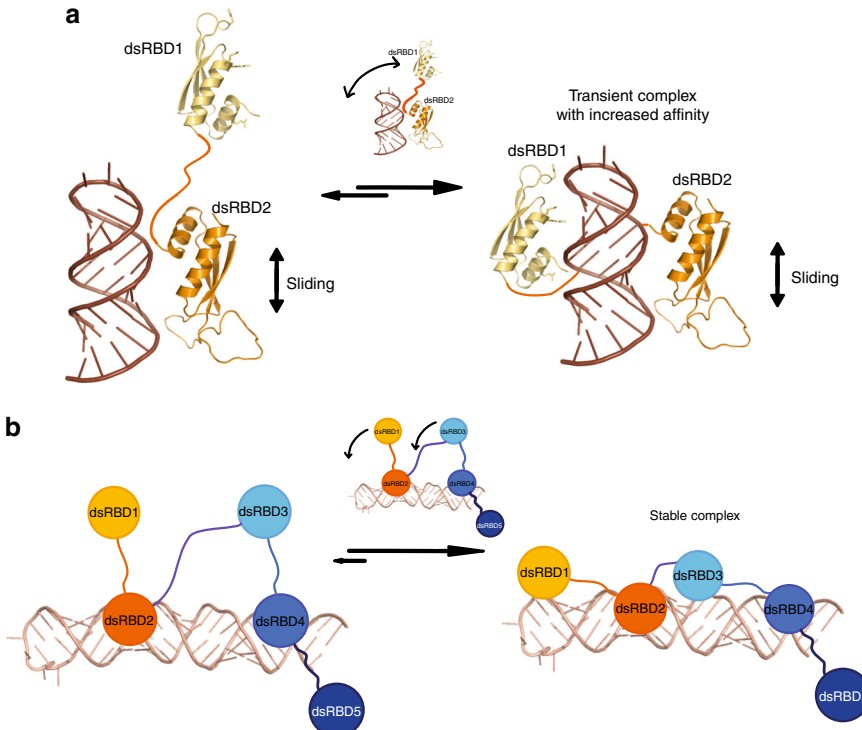

**Fig. 7** Model of the molecular recognition of dsRNA by mStau2. **a** dsRNA recognition by the mStau2 tandem domain dsRBD1–2. dsRBD2 binds dsRNA promiscuously with moderate affinity and slides along the stem. Through this sliding, dsRBD2 positions dsRBD1 close to the dsRNA. When a suitable dsRNA structure is reached, dsRBD1 also binds, thereby strongly increasing the affinity of the tandem domain to dsRNA. **b** Next to the tandem domain dsRBD 1–2, dsRBD 3–4 acts in a similar way. Here, dsRBD4 does the first promiscuous binding with moderate affinity. When dsRBD2 and dsRBD4 position dsRBD1 and dsRBD3, respectively, close to a suitable dsRNA, the respective domains also bind the dsRNA, thereby increasing affinity. Only when suitable dsRNA binding sites for both tandem domains are in sufficient spatial proximity can all four dsRBDs be bound and form a stable complex with the RNA target

mStau2. However, for most of these multidomain proteins the manner in which they act cooperatively for function and specificity is not well understood. mStau2 contains two tandem domains, each of which can bind to secondary structures. It is likely that the combination of secondary structure elements as well as their spatial arrangement determine the specificity of Stau binding for transport of selected mRNAs in vivo.

## Methods

**Molecular cloning**. DNA sequences of interest were amplified by polymerase chain reaction (PCR) from template plasmids. Cloning was performed with the In-Fusion HD Cloning kit (Clontech) according to the manufacturer's protocol. Point

mutations or deletions were introduced by 3-point PCR with overlap extension[35] or with the QuikChange II Site-directed mutagenesis kit (Agilent technologies), according to the manufacturer's instructions.

For in vivo experiments in *D. melanogaster*, rsEGFP2 was fused with mStau2, via cloning into pBlueScript-KS. First, the primers pBSKS-rsEGFP2 FW and rsEGFP + 3 C RV were used to create an rsEGFP2 sequence with a pBlueScript-KS 5′ overhang for In-Fusion cloning and a PreScission protease cleavage site as 3′ overhang, and the primers 3 C + Stau2 FW and pBSKS-mStau2 RV to create a mStau2 sequence, with a PreScission cleavage site 5′ overhang and a pBlueScript-KS 3′ overhang for In-Fusion cloning. The PCR products from these reactions served as templates for a third PCR with the primers pBSKS-rsEGFP2 FW and pBSKS-mStau2 RV to create rsEGFP2-mStau2 sequences with 5′ and 3′ overhangs for In-Fusion cloning into BamHI/XbaI-linearized pBlueScript-KS, according to the manufacturer's protocol. The resulting plasmids served as templates for PCR

with primers pUASp-rsEGFP2 FW and pUASp-mStau2 RV or pUASp-dmStau RV to amplify rsEGFP2-mStau2/dmStau sequences with 5′ and 3′ overhangs for In-Fusion cloning into the BamHI/XbaI-linearized pUASp-attB plasmid.

For details on plasmids and primer sequences, see Supplementary Tables 2, 3, and 4.

**Expression of full-length mStau2 protein**. mStau2 FL was expressed as a HisSUMO-tagged fusion protein in High Five insect cells. After cloning in pFastBacDual, recombinant baculovirus was produced with the Bac-to-Bac Expression System (Invitrogen) in Sf21 insect cells as described by the manufacturer's protocol.

**Expression of truncated mStau2 dsRBD protein constructs**. mStau2 proteins were expressed after cloning into the expression vector pOPINS3C as fusion proteins with HisSUMO-tag in E. coli Rosetta cells using autoinduction ZY-medium[36].

**Expression of isotope-labeled proteins for NMR**. Uniformly $^{15}$N- or $^{15}$N,$^{13}$C-labeled proteins for NMR experiments were expressed in $^{15}$N-M9 minimal medium (1 × $^{15}$N-labeled M9 salt solution, 0.2% ($^{13}$C-) glucose, 1 mM MgSO$_4$, 0.3 mM CaCl$_2$, 1 μg per L biotin, 1 μg per L thiamine, 1× trace metals) supplemented with antibiotics. Hundred milliliters precultures were grown overnight at 37 °C, shaking at 150 rpm and used to inoculate 1 L prewarmed M9 minimal medium. Cultures were grown at 37 °C, 150 rpm to OD$_{600nm}$ = 0.6. Protein expression was induced with 0.25 mM IPTG, and cultures were cooled for protein expression overnight at 18 °C.

**Purification of full-length mStau2 protein**. High Five cell pellets containing HisSUMO-tagged mStau2 FL were lysed by sonication in lysis buffer (1× PBS, 880 mM NaCl, 400 mM arginine, 2 mM DTT, and 10 mM imidazole). The lysate was cleared by centrifugation, and the soluble protein fraction was purified by Ni-IMAC on HisTrap FF (GE). Bound protein was eluted with 200 mM imidazole after extensive washing with 15 CV lysis buffer. The protein was dialyzed in low salt buffer (40 mM Bis-Tris pH 7, 150 mM NaCl, 50 mM arginine, 2 mM DTT) overnight before further purification on a HiTrap Heparin HP column (GE) and size exclusion chromatography (SEC) on Superdex200 Increase (GE).

**Purification of truncated mStau2 dsRBD protein constructs**. E. coli Rosetta cell pellets containing HisSUMO-tagged fusion proteins were lysed by sonication in lysis buffer (1× PBS, 880 mM NaCl, 400 mM arginine, 2 mM DTT, 10 mM imidazole). The lysate was cleared by the centrifugation, and the soluble protein fraction was purified by Ni-IMAC on HisTrap FF (GE). Bound protein was eluted with 200 mM imidazole after extensive washing with 15 CV lysis buffer. For fusion-tag removal, the protein was digested overnight with PreScission protease upon dialysis in low-salt buffer (40 mM Bis-Tris pH 7, 150 mM NaCl, 50 mM arginine, 2 mM DTT). The protein was purified with a second, subtractive Ni-IMAC affinity chromatography, on a HiTrap Heparin HP column (GE) and size exclusion chromatography (SEC) on Superdex 75 (GE). Size-exclusion chromatography was performed in minimal buffer (40 mM Bis-Tris pH 7, 150 mM NaCl, 2 mM DTT) or the indicated buffer required for downstream applications.

**Small-scale RNA in vitro transcriptions**. RNAs for EMSAs were produced by small scale in vitro transcriptions with the MegaShortScript T7 Transcription kit (Ambion) according to the manufacturer's protocol. HPLC-purified primers (Eurofins) were used as templates (Supplementary Tables 1 and 5). In order to produce partially double-stranded template DNA, FW (T7prom) and RV primers were annealed after unfolding at 60 °C for 5 min by slow cooling to RT.

**Large-scale RNA in vitro transcriptions**. SRS2 and SRS2 + 5Δloop RNAs were purchased from IBA (Göttingen). Other RNAs needed in large amounts for NMR experiments were produced by large-scale in vitro transcription. As a template, 4 μM HPLC-purified FW (T7prom) primer and 3.4 μM HPLC-purified RV primer (Supplementary Tables 1, 5) were annealed after unfolding at 60 °C for 5 min by slow cooling to RT in 34 mM MgCl$_2$ in a total volume of 594 μL. This DNA template mixture was used for a 5 mL in vitro transcription reaction containing, in addition to the template, 4 mM of each NTP, a template specifically optimized concentration of MgCl$_2$ (see below), 80 mg per mL PEG8000 and 0.5 mg per mL T7 RNA polymerase in 1× TRX buffer (40 mM Tris/HCl pH$_{RT}$ 8.0, 1 mM spermidine, 0.1‰ Triton X-100, 5 mM DTT). The reaction was incubated for 2 h at 37 °C. The reaction was stopped by removal of precipitants by centrifugation at 48,384 g for 5 min and subsequent RNA precipitation with 0.1 V 3 M NaOAc and 3 V absolute ethanol at −20 °C overnight.

The optimal MgCl$_2$ concentration for each RNA was determined beforehand by MgCl$_2$ screening in 50 μL reactions containing 4–60 mM MgCl$_2$. Quality and quantity of RNA in each MgCl$_2$ concentration were examined by 8% urea PAGE.

**PAGE purification of RNA**. RNA was pelleted by centrifugation at 48,384 g, 4 °C, for 30 min, air-dried and subsequently dissolved in 1× denaturing RNA loading dye. The RNA was purified by 8% 1x TBE- 8 M urea PAGE in an Owl sequencing chamber (Thermo Fisher Scientific) in 1x TBE running buffer at constant 300 V for 17–20 h. RNA bands were visualized by UV shadowing and the desired band was excised from the gel and extracted by electroelution in a Whatman Elutrap electroelution system (GE Healthcare) at constant 200 V in 1x TBE for 8 h. Eluted RNA was collected each hour. Eluted RNA was dialyzed against 5 M NaCl at 4 °C overnight and subsequently twice against RNase-free water at 4 °C overnight before drying in a Concentrator Plus SpeedVac (Eppendorf).

**Radioactive labeling of RNA**. RNAs for electrophoretic mobility shift assays (EMSA) were labeled radioactively for sensitive detection of protein–RNA interactions. In vitro transcribed RNA was 5′ dephosphorylated in 20 μL reactions containing 10 pmol RNA, 1x Tango buffer with BSA (Thermo Fisher), 2 U FastAP thermosensitive alkaline phosphatase (Thermo Fisher) and 20 U of the RNase inhibitor SUPERaseIn (Thermo Fisher). After incubation at 37 °C for 15 min, the dephosphorylated RNA was phenol/chloroform extracted and precipitated with 0.1 V 3 M NaOAc, 3 V absolute ethanol and subsequent chilling at −20 °C for ≥15 min.

For radioactive labeling, 10 pmol dephosphorylated RNA or chemically synthesized RNA were 5′-phosphorylated with $^{32}$P from γ-$^{32}$P ATP (Hartmann Analytic) in a 20 μL reaction with T4 polynucleotide kinase (New England Biolabs) in 1× buffer A. The labeling reaction was incubated at 37 °C for 30 min and subsequently stopped at 72 °C for 10 min.

Remaining free nucleotides were removed by purification on a NucAway™ Spin column (Ambion) according to the manufacturer's instructions. Eluted radiolabeled RNA was diluted to a final concentration of 100 nM in RNase-free H$_2$O and stored at −20 °C.

**Electrophoretic mobility shift assay (EMSA)**. For EMSAs with short RNAs (<100 nt), protein at the indicated final concentration was mixed with 5 nM radiolabeled RNA in RNase-free protein buffer supplemented with 4% glycerol and 30 μg per mL yeast tRNA as a competitor in a final volume of 20 μL. In order to allow protein–RNA complexes to form, the mixtures were incubated for >20 min at RT.

Separation of protein–RNA complexes was performed by native PAGE on 6% polyacrylamide 1x TBE gels in 40 min at constant 110 V in 1x TBE running buffer. Subsequently, the gels were fixed in 30% (v/v) methanol, 10% (v/v) acetic acid for 10 min before drying in a vacuum gel drier (BioRad). Visualization of radioactivity occurred after exposure of radiograph films (Kodak) in a Protec Optimax developer (Hohmann) or by PhosphorImaging with a Fujifilm FLA-3000. Each experiment was performed as a triplicate on different days.

In the case of long, unlabeled RNA (>100 nt), 10–100 nM RNAs were used, and separation of protein–RNA complexes was performed by 1–1.5% agarose gel electrophoresis. Visualization of RNA was achieved by GelRed (Biotium) staining. Fluorescence was visualized with a Fusion SL imaging system (Vilber Lourmat) by UV at 254 nm.

**Biotinylation of RNA**. Ligand RNA for binding studies by Surface Plasmon Resonance was biotinylated to allow immobilization on a streptavidin-coated surface. For biotinylation, the Pierce™ RNA 3′ End Biotinylation Kit (Thermo Fisher Scientific) was used according to the manufacturer's protocol. Fifty pmol of RNAs were used per 30 μL reaction. After extraction and precipitation, the RNA was redissolved in 100 μL RNase-free water.

**Surface plasmon resonance**. Surface Plasmon Resonance (SPR) experiments were performed with a BIACORE 3000 system (GE Healthcare). To assess protein–RNA interactions, biotinylated RNA in a volume of 60 μL was streptavidin-captured on a SA-Chip (GE Healthcare) surface at a flow rate of 10 μL per min after three consecutive 1 min conditioning injections of 50 mM NaOH, 1 M NaCl.

Full-length mStau interacted strongly with the blank SA-Chip surface. Thus, to assess the interaction of RNA with mStau FL, the protein was diluted in HBS-EP buffer (10 mM HEPES pH 7.4, 150 mM NaCl, 3 mM EDTA, 0.005% surfactant P20) and covalently amine-coupled to a CM5-Chip (GE Healthcare) according to the manufacturer's instructions.

Kinetic analysis of protein–RNA interactions was performed at a flow rate of 30 μL per min in HBS-EP buffer. Analyte protein or, in case of mStau2 FL, RNA in HBS-EP buffer at the indicated concentrations was injected for 4–5 min to allow for association, subsequent dissociation was allowed for 10 to 15 min in HBS-EP buffer. To remove any residual bound protein, two 1 min regeneration injections with 1 M NaCl were performed.

Data were analyzed in the BIAevaluation software (GE Healthcare). After double-referencing of obtained binding curves against the signal in a ligand-free reference channel or, in the case of full-length mStau2, against a HisSUMO-coupled reference channel, and a buffer run, average values for the analyte response at equilibrium were calculated. Steady-state binding curves were obtained by plotting the response at equilibrium against analyte concentration and curve fitting

with the Origin Pro 9.0 (OriginLab) software using the two-site binding or Hill1 fits available in the software. All experiments were performed at least in triplicate on different days and the results of the $n \geq 3$ experiments were averaged. Kinetic fits, if applicable, were performed with the BIAevaluation software (GE Healthcare) using the bivalent analyte model available in the software.

**Circular dichroism spectroscopy.** Circular dichroism (CD) spectra were collected with a Jasco J-715 spectropolarimeter from 260 nm to 190 nm in a continuous scanning mode with a scanning speed of 50 nm per min at a bandwidth of 1 nm. Five scans were collected per measurement and response time was 8 s. The measurements were performed at 20 °C. Spectra were analyzed with SpectraManager (Jasco).

**Static light scattering.** Static light scattering (SLS) experiments were performed with a 270 Dual Detector and a VE3580 RI Detector (Malvern) after SEC on a 10/300 Superdex200 Increase column (GE Healthcare) at 7 °C and a flow rate of 0.5 mL per min. System calibration was performed with BSA. Data were analyzed with the OmniSEC 5.02 software (Malvern).

**Structure determination of SRS2 RNA.** Crystallization experiments were performed at the X-ray Crystallography Platform at Helmholtz Zentrum München. Initial crystals of *Rgs4* SRS2 were obtained in 80 mM NaCl or 80 mM KCl, 20 mM $BaCl_2$, 40 mM Na cacodylate pH 6.5, 40% MPD, 12 mM spermine after 4 days at room temperature using the sitting-drop vapor-diffusion method. Crystals could be reproduced in 34–46% MPD and were harvested with cryogenic loops and flash-frozen in liquid nitrogen. Native datasets were collected at the Swiss Light Source (SLS) synchrotron, beamline PXIII. Anomalous data for phasing were collected at the European Synchrotron radiation Facility (ESRF) at beamline ID 23–2. Data were indexed and integrated using XDS and scaled via XSCALE. Structure factor amplitudes were obtained with Truncate (CCP4 package)[37]. The structure was solved by MAD phasing with Barium from the mother liquor, using the Auto-Rickshaw web server[38]. The structure was completed by iterative manual building in COOT and refinement with RefMac5 (CCP4 package)[37]. All crystallographic software was used from the SBGRID software bundle. Images of the crystal structure were prepared with PyMol (Version 1.7; Schrodinger; http://www.pymol.org/) or CueMol (Version 2.2; http://www.cuemol.org/en/).

**Nuclear magnetic resonance (NMR).** RNA and RNA–protein complexes were dialyzed to 150 mM NaCl, 20 mM sodium phosphate buffer, pH 7.0, 5 mM DTT prior to analysis, and 5–10% $D_2O$ was added for locking. Measurements were performed at 298 K on Bruker AVIII 600, AVIII 800 or AVIII 900 NMR spectrometers with cryogenic (TCI) triple resonance gradient probes. Data were processed with Topspin 3.0 or Topspin 3.5 and analyzed with Sparky 3[39] and CcpNmr Analysis[40]. RNA assignment of imino groups was based on $^1H,^1H$-NOESY spectra; an initial protein backbone assignment was made with HNCACB spectra. Titration experiments with the single mStau2 dsRBDs 1 and 2 as well as the tandem domain dsRBD1–2 were performed at 50 µM protein concentration. After snap-cooling, the RNA ligand was added in molar ratios of 0.5, 1.0, 2.0, and 3.0 to the protein.

To determine the binding interface of Stau2 dsRBD1–2 to RNA, imino signals of the unbound RNA were compared with the respective resonances of a Stau2 dsRBD1–2-RNA (1:1) complex. The bound spectrum was scaled so that imino signals, which do not show additional exchange-mediated line broadening in the complex, have the same peak height as in the free spectrum (still with larger line broadening).

**Fly strains and transgenesis.** Heterozygous combination of the stau[D3][41] (FBal0016165) and stau[R9][5] (FBal0032815) alleles was used to generate females lacking endogenous *dmstaufen*. The αTub67C:GFP[m6]-Staufen[42] (FBal0091177) transgene was used as a source of GFP-dmStau in the female germline. Expression of the UASp-GFP-mStau2 transgenes was driven with one copy of oskar-Gal4[43] (FBtp0083699) in the female germ line. $w^{1118}$ (FBal0018186) was used as the wild-type control. All UASp-GFP-mStau2 transgenes were generated by subcloning wild-type or mutant mStau2 coding sequences 3′ to the cds of rsEGFP2 into the pUASp-attB trangenesis vector[44]. The UASp-rsEGFP2-mStau2 vectors were injected into embryos of y[1] M{vas-int.Dm}ZH-2A w[*]; PBac{y[ + ]-attP-3B} VK00033 (FBti0076453) females to facilitate psiC31 mediated insertion into the same locus on the 3 L chromosome arm. All stocks were raised on normal corn-meal agar at 25 °C.

**Single molecule fluorescent hybridization (smFISH).** Forty-two and 24 different ssDNA oligonucleotides were labeled enzymatically with Atto532-ddUTP and Atto633-ddUTP, respectively, as described in refs. [45,46], to generate osk42x532[45] and bcd24x633 (Supplementary Table 6) probe-sets for smFISH. Briefly, 1000 pmol of manually selected, non-overlapping arrays of desalted DNA oligos (Sigma-Aldrich GmbH) complementary to *oskar* or *bicoid*, respectively, were mixed with labeled ddUTPs and 0.006 U per pmol TdT enzyme in 1x TdT buffer and

incubated at 37 °C overnight[45,47]. *Drosophila* ovaries were dissected into 2% v/v PFA, 0.05% Triton X-100 in PBS and fixed for 20 min. The ovaries were washed twice in PBT (PBS + 0.1% (v/v) Triton X-100, pH 7.4) for 5 min. Ovaries were prehybridized in 200 µL 2 × HYBEC (300 mM NaCl, 30 mM sodium citrate pH 7.0, 15% (v/v) ethylene carbonate, 1 mM EDTA, 50 µg per mL heparin, 100 µg per mL salmon sperm DNA, 1% (v/v) Triton X-100) for 10 min at 42 °C. In total, 50 µL prewarmed probe mixture (5 nM per individual oligonucleotide) was added to the prehybridization mixture and hybridization was allowed to proceed for 2 h at 42 °C. Free probe molecules were washed out of the specimen by two washes with prewarmed HYBEC and a final wash with PBT at room temperature. Ovaries were mounted in Vectashield and processed for smFISH analysis. For further details, see refs. [45,46].

**Microscopy and image analysis.** *Drosophila* egg-chambers mounted onto slides in Vectashield were imaged on a Leica TCS SP8 confocal laser scanning microscope using a 20x dry (NA = 0.75) objective for imaging the RNA distribution and a 63x oil immersion (NA = 1.4) objective to obtain high-resolution images for co-localization analysis of *oskar* or *bicoid* mRNA and GFP-Stau. The outlines of the oocytes and the anteroposterior (AP) axis were manually specified, and the smFISH signal was redistributed into a 100 × 100 matrix. Each column of this matrix represents the relative amount of signal found under 1% of the AP axis length with anterior on the left (column 1) and posterior on the right (column 100). Such matrices are then averaged to obtain a mean and the variability of the RNA localization during a certain stage of oogenesis. Moreover, descriptors such as the center of mass (relative to the geometric center of the oocytye) and the amount of RNA localizing to the posterior domain (defined on the minimum two-fold enrichment of the signal over what is expected) were extracted and compared statistically using a Kruskal–Wallis test followed by pairwise Mann–Whitney U test against the *stau* null condition. More details on the analyses of RNA distribution within stage 9 and stages 10–11 oocytes are described in ref. [48].

Colocalization between the mRNAs and GFP-Stau was assayed by image deconvolution using Huygens essential segmentation and establishment of nearest neighbor pairs between *oskar* or *bicoid* mRNPs and GFP-Stau particles[49]. To determine the number of mRNA molecules in an mRNP and to normalize GFP-Stau signal intensity, we fitted multiple Gaussian functions to the corresponding signal intensity distributions taken from the nurse cells using the mixtools package in R (https://cran.r-project.org/web/packages/mixtools/index.html)[45,50]. The µ value of Gaussian fit that described the largest portion of the distribution (for *oskar* mRNPs ~ 60%, for *bicoid* mRNPs ~ 80%, GFP-Stau >85%) was taken as the signal intensity of a unit (for mRNPs the intensity of a signal mRNA molecule). The chosen µ value was always the smallest among the µ values of the fitted Gaussians. Raw signal intensities were normalized with the determined unit values. RNPs were clustered based on this normalized intensity under the following rule: $[2^i:2^{i+1}]$, i ∈ [0:8], i.e., 1, 2:3, 4:7, 8:15, etc. The observed nearest neighbor colocalization frequencies were computed for each of the clusters and were compared to the expected co-localization frequencies (governed by the object-densities, determined in randomized object localizations[49]. Linear correlation between RNA content and GFP-Stau intensity was established ($R^2 > 0.9$ in all cases, except between [F40A F157A] and bicoid, $R^2 = 0.61$). The slopes of the fitted lines were compared pairwise using least-squares means analysis[51]. All statistical analyses were carried out in R[52] using RStudio (www.rstudio.com). All graphs were plotted by the ggplot2 library in R[53].

**Reporting summary.** Further information on experimental design is available in the Nature Research Reporting Summary linked to this article.

## Data availability

Source data are available for Figs. 1, 2, 4, 5, and 6. All structural data are available at the Protein Databank, https://www.rcsb.org, with accession number 6H0R. The data that support the findings of this study are available from the corresponding author upon reasonable request.

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

## Acknowledgements

We thank Vera Roman for technical support, Gregor Witte for support with static light-scattering experiments, and Michael A. Kiebler for fruitful discussions and support. We acknowledge the support by the Bavarian NMR center and the X-ray crystallography platform at the Helmholtz Zentrum München. X-ray data were collected at the Swiss Light Source (SLS) synchrotron, beamline PXIII and the European Synchrotron radiation Facility (ESRF) at beamline ID 23–2. This work was supported by the Deutsche Forschungsgemeinschaft (FOR2333 to S.H., I.G., A.E., and D.N.).

## Author contributions

S.H., I.G., A.E., M.S., and D.N. conceived and designed the experiments. S.H., I.G., J.-N. T., J.G., S.M.F.M., and R.J. performed the experiments. S.H., I.G., R.J., J.-N. T., J.G., A.E., M.S., and D.N. analyzed the data. S.H. and D.N. wrote the paper. All authors reviewed and approved the paper.

## Additional information

**Competing interests:** The authors declare no competing interests.

