## [Peer Review File · Nature Communications]

Reviewers' comments:

Reviewer #1 (Remarks to the Author):

In this manuscript, Heber et al biochemically, structurally and functionally characterize the RNA-binding properties of mouse Staufen 2 (mStau2) with a focus on the first two dsRNA binding domains (dsRBDs). These domains were previously suggested to not bind dsRNA, however, the authors use a combination of NMR, SPR and EMSA experiments to show that dsRBD1 and 2 bind dsRNA with affinities and kinetics, similar to the third and fourth dsRBDs. In order to determine if dsRBD1+2 can contribute to recognition of mRNAs in vivo, the authors use staufen-deficient *Drosophila* oocytes and measure the rescue of oskar mRNA localization to the oocyte posterior. While mStau2 can partially restore oskar mRNA localization, the mStau2 dsRBP1+2 mutants cannot. Understanding the binding specificity of the Staufen-family of proteins has been a major challenge for the field because of its multiple dsRBPs and this work helps to clarify the function of these domains. The experiments are performed and described well and the insights are an important contribution to the field.

Comments:

1. The work in this manuscript focuses on dissecting the interaction of mStau2 with the Rgs4 3'UTR because it was previously identified as a target of Stau2. It is not clear to me, however, if there is truly specificity for this transcript. Would similar results be obtained with any dsRNA over a certain length? While it appears from the experiments that dsRBD2 and dsRBD4 provide general dsRNA binding recognition, could dsRBP1 and dsRBP3 provide some specificity that was not within the SRS1 and SRS2 sequences? I appreciate the difficulty in identifying bona fide targets of mStau2, so I think some discussion of the interpretation of the specificity would be helpful
2. My understanding of the previous structural work on isolated dsRBDs is that this domain recognizes the shape of A-form dsRNA and interacts with two successive minor grooves. For this reason, many of the RNAs used in this work seem to be too short to accommodate multiple dsRBDs. Is this the reason why the SRS2+5 RNA binds more tightly?
3. Line 496-498 It is not intuitive to me why the fast kinetics observed for Kd1 would only allow Kd2 to be observed in EMSA. Shouldn't the kinetics of the weaker complex be even faster. Is it possible that the complex is just not stable in EMSA and that it disassociates while running the gel? It is not clear to me that this is actually a stable complex arising from Kd2.
4. line 506-508 I believe a Hill Coeff greater than 1 indicates cooperativity between multiple mStau2 molecules binding the same RNA and not the interactions between domains of a mStau2 monomer. Throughout the text, the term cooperativity is used loosely. I would reserve the word cooperative only for its strict definition.
5. line 536 I find the phrase simultaneously and rather independently to be slightly confusing and I am not sure exactly what they are indicating
6. line 612-613 I do not understand how the mutation can be compensated for with a longer RNA without additional interactions being made, but it is not clear to me what where these contacts are coming from.
7. line 651-652 Why does the double mutant in dsRBD1+2 affect the binding of dsRBD3+4 in the dsRBP1-4 construct? Shouldn't it bind the same as the isolated dsRBD3+4 domain?
8. Does the partial rescue of oskar mRNA localization by Stau2 indicate that there are some differences in specificity between the two proteins? Can the basis of this be explained or discussed? To me, this suggests that some additional sequence determinants must still exist, but I think that is beyond the scope of this manuscript to define them.

Minor comments:

1. Figure S1: Figure legend for the schematic is missing and I would move this to Fig 1 since I think this information is helpful for interpreting the experiments throughout the paper

2. Figure S3: Electron density map (2Fo-Fc?) and sigma contour should be stated
3. Figure S8: a mapping of the mutations onto a dsRBD structure would be helpful
4. Figure S9: Referenced out of order in the text
5. Figure S10-14: I find the star marking the position of the mutation in the dsRBD domain to be a little confusing because a single star is used to denote multiple different mutations in the schematic. I would try to clarify this with labeling of the mutation or the use of different symbols. Also labeling in panels and figure legend are not consistent in Fig S10, S11, .
6. line 539 should refer to figure 1F and not 1E

Reviewer #2 (Remarks to the Author):

This article describes biochemical and functional studies of the murine protein Staufe2, in particular with respect to the mRNA binding capacity of its two domains dsRBDs 1 and 2. These domains had been reported previously not to bind RNA, while the RNA-binding capacity of Staufe2 had been attributed exclusively to dsRBDs 3 and 4. Using EMSA, SPR and NMR, the authors demonstrate that dsRBDs 1-2 also contribute to RNA binding. This conclusion is fully supported by the experiments and unquestionable. However, I have many problems with the quantitative interpretation of nearly all experiments and cannot recommend publication of this article in the present form. In details:

1. The EMSAs of Figure 1 show that SRS2, and SRS2+5 are shifted in two distinct bands depending on the protein concentration by both dsRBDs 1-2 and dsRBDs 3-4. This "double shift" cannot be recapitulated by the subsequent binding of the two dsRBDs present in each construct, as an intermolecular binding of the second domain of the tandem dsRBDs, following the binding of the first domain, should not depend on the concentration of the protein. The authors do not comment on this.
2. On line 459, the line broadening of the NMR signals is interpreted in terms of sliding of the protein on the RNA. This is not in agreement with the localization of the line broadening to the last four base pairs (line 436-437). If the protein slides along the long stem of SRS2+5, thus contributing an entropic factor to the binding energy, and if this sliding translates into imino line broadening, why should 4 base pairs at the terminus be preferentially affected?
3. A 7 base pair stem is considered to be the minimum sequence that binds the tandem dsRBDs. However, the SPR experiments were performed on a much longer sequence, which then potentially contains multiple binding sites. I would believe the conclusions from the following SPR experiments much more if they had been performed on this minimal construct rather than on the longer SRS* or SRS2+5 constructs.
4. The binding curves of dsRBDs 1-2 and dsRBDs 3-4 to the SRS* and SRS2+5 RNAs show a two-site binding behavior. Later (line 710-713), it is suggested that these two events correspond to RNA binding of the first and the second dsRBD in the tandem construct. This is surprising, as binding of tandem domains to a tandem substrate (the RNA) should not depend on protein concentration. The local concentration of the second binding event depends only on the saturation of the first binding event. These two binding events may represent the binding of two copies of dsRBDs 1-2 or dsRBDs3-4 to the RNA. The meaning of the two binding events is not discussed in the article and the suggestion of line 710-713 is in my opinion wrong. The same is true for the interpretation of the Hill coefficient of 1.7 as a cooperative behavior of the four domains belonging to the same protein. What excludes some "cooperativity" between two protein copies? In addition, interpreting the Hill coefficient as a sign of cooperativity is an oversimplification of more complicated, case-specific models. The authors should apply these models to interpret the binding curve, considering that the local concentration of the second, third and fourth dsRBD is influenced by binding of the first one. This might be very complicated; on the other hand a wrong over simplification is not of much help.

5. In the NMR analysis of isolated dsRBD1 and dsRBD2 domain it is noticed that dsRBD1 is present in several different conformations. From the text it remains unclear whether this conformational multiplicity is present also in the tandem dsRBDs 1-2 construct. Inspection of Fig. S6 and comparison with Figure 2 shows that this is indeed the case. Nevertheless, on line 539 it is written "Interestingly, for dsRBD1 less line-broadening is observed for imino signals at a protein:RNA ratio of 1:0.5. This indicates that the two dsRBDs bind the RNA with different dynamics and possibly reduced affinity...." I am surprised of such an observation. The authors seem to forget that, if the protein is present in different conformations, only one of these conformations is competent for binding (as shown also in Fig. 2A), thereby reducing the effective protein concentration!!!! The observation that differential line broadening at the same RNA:protein ratio indicates different dynamics is, in this context, simply wrong.
6. On the same page, the RNA binding activity of dsRBD1 and dsRBD2 are compared on line 545-573. This paragraph ignores that the multiple conformations observed for dsRBD1 may be an artifact of the truncated construct. The presence of these multiple conformations diminishes the binding-competent concentration of protein. Thus, it remains unclear whether the very weak binding activity of dsRBD1 is a property of dsRBD1 in the context of the full protein.
7. Discussion of dsRBD1 mutations, line 591 to 613. Which should be the mechanism by which the length of the RNA compensates a mutation in the canonical RNA binding site of a dsRBD?
8. The dsRBD2 mutation H169A has a more dramatic effect on RNA binding than F157A when tested in the dsRBDs 1-2 construct. However, this mutation becomes less deleterious than F157A when seen in the context of the dsRBDs 1-4 and together with the F40A mutation. Why?
9. The ability of mStau2 to complement DmStau is only moderate. In light of this, I wonder how significant the functional assays of Fig. 6 are. Clearly, this moderate complementation ability may indicate that the mode of recognition of mStau2 versus the DmRNA might differ from the native one. In this context, what are the experiments with the mutant mStau2 telling us?

Reviewer #1 (Remarks to the Author)

In this manuscript, Heber et al biochemically, structurally and functionally

characterize the RNA-binding properties of mouse Staufen 2 (mStau2) with a focus

on the first two dsRNA binding domains (dsRBDs). These domains were previously suggested to not bind dsRNA, however, the authors use a combination of NMR, SPR

and EMSA experiments to show that dsRBD1 and 2 bind dsRNA with affinities and kinetics, similar to the third and fourth dsRBDs. In order to determine if dsRBD1+2 can contribute to recognition of mRNAs in vivo, the authors use staufen-deficient *Drosophila* oocytes and measure the rescue of oskar mRNA localization to the oocyte posterior. While mStau2 can partially restore oskar mRNA localization, the mStau2 dsRBP1+2 mutants cannot. Understanding the binding specificity of the Staufen-family of proteins has been a major challenge for the field because of its multiple dsRBPs and this work helps to clarify the function of these domains. The experiments are performed and described well and the insights are an important contribution to the field.

Comments:

1. The work in this manuscript focuses on dissecting the interaction of mStua2 with

the Rgs4 3'UTR because it was previously identified as a target of Stau2. It is not

clear to me, however, if there is truly specificity for this transcript. Would similar

results be obtained with any dsRNA over a certain length? While it appears from the experiments that dsRBD2 and dsRBD4 provide general dsRNA binding recognition,

could dsRBP1 and dsRBP3 provide some specificity that was not within the SRS 1 and SRS2 sequences? I appreciate the difficulty in identifying bona

fide targets of

mStau2, so I think some discussion of the interpretation of the specificity would be

helpful

We fully agree with this interpretation and do believe that similar results would be obtained with any dsRNA of similar length. While we cannot exclude specificity of dsRBDs for certain sequence motifs, we found no experimental evidence for such an assumption. This is in line with previous observations for canonical dsRBDs, which also indicate interactions with the RNA backbone. We thus believe that indeed the Stau2 protein recognizes its RNA target in a structure- and not sequence-dependent manner.

In the Discussion we added the following statement (lines 654-660): *“While we cannot exclude specificity of dsRBDs for certain sequence motifs, we found no experimental evidence for such an assumption. Our data rather suggest that the Stau2 protein recognizes its RNA target in a structure- and not sequence-dependent manner. Likely scenarios for the specificity reported in vivo include the recognition of combinations of secondary structure elements or a contribution of cofactors.”*

2. My understanding of the previous structural work on isolated dsRBDs is that this domain recognizes the shape of A-form dsRNA and interacts with two successive minor grooves. For this reason, many of the RNAs used in this work seem to be too short to accommodate multiple dsRBDs. Is this the reason why the SRS2+5 RNA binds more tightly?

This is indeed a possible reason for higher affinity binding by the longer RNA. We appreciate this statement as this was a major reason why we did not use the minimal 7-basepair stem loop, as requested by reviewer 2, but rather the longer SRS2+5 stem loop RNA. We also used a stem-loop that had been previously described as likely binding site for Staufen proteins (SRS2; Laver et. Al., 2013 and Heraud-Farlow et. al., 2013) and thus is likely to have physiologic length. We believe that the combination of these analyses yields a better understanding of mStau2 binding than just the minimal 7-basepair stem loop, the SRS2, SRS*, or SRS2+5 alone.

For Biacore experiments, we decided to use SRS2+5 and SRS* RNAs because we reasoned that their additional lengths prevent steric hindrance from the dextran surface of the Biacore chips.

3. Line 496-498 It is not intuitive to me why the fast kinetics observed for Kd1 would only allow Kd2 to be observed in EMSA. Shouldn't the kinetics of the weaker

complex be even faster. Is it possible that the complex is just not stable in EMSA and that it disassociates while running the gel? It is not clear to me that this is actually a stable complex arising from K_D2.

We agree that the transient complex most likely dissociates during electrophoretic separation on the gel. However, at rather high micromolar concentrations of the protein, we observe a gel shift in EMSA, indicating that at high, non-physiological concentrations, the complex remains associated. We can, however, indeed not proof that this is the binding with K_D2 we observe by SPR.

In order to account for this ambiguity, we decided to remove this statement from the manuscript.

4. line 506-508 I believe a Hill Coeff greater than 1 indicates cooperativity between multiple mStau2 molecules binding the same RNA and not the interactions between domains of a mStau2 monomer. Throughout the text, the term cooperativity is used loosely. I would reserve the word cooperative only for its strict definition.

To us it is unclear why a Hill coefficient >1 would only occur in intermolecular and not in intramolecular interactions. Both options should be possible. However, in order to account for this concern, we removed the word “cooperativity” throughout the manuscript whenever the case was ambiguous.

5. line 536 I find the phrase simultaneously and rather independently to be slightly confusing and I am not sure exactly what they are indicating

We changed the sentence in the following way (lines 316-319): “...*suggesting that both domains bind the RNA ~~simultaneously and rather independently~~*”

6. line 612-613 I do not understand how the mutation can be compensated for with a longer RNA without additional interactions being made, but it is not clear to me what where these contacts are coming from.

We agree that this statement is not well justified and thus removed it from the text.

7. line 651-652 Why does the double mutant in dsRBD1+2 affect the binding of dsRBD3+4 in the dsRBP1-4 construct? Shouldn't it bind the same as the isolated dsRBD3+4 domain?

Indeed, the results with the mutated dsRBD1-2 F40A F157 indicate that the RNA binding is not fully abolished and rather slows down the on-rates. This is consistent with the observation that the double mutation F40A F157 in the context of dsRBD1-2 only has a similar effect (Supplementary Figure 15). Considering that the RNA binding is not fully abolished, in our eyes the results with the construct dsRBD1-4 F40A F157 is therefore rather expected.

We would like to point out, however, that RNA binding in case of dsRBD1-4 F40A H169A behaves similar to wild-type dsRBD3-4 alone.

8. Does the partial rescue of oskar mRNA localization by Stau2 indicate that there are some differences in specificity between the two proteins? Can the basis of this be explained or discussed? To me, this suggests that some additional sequence determinants must still exist, but I think that is beyond the scope of this manuscript to define them.

We agree that this is possible and quite likely. We indeed plan to follow up on this observation in our future work. We also added the following statement in the Discussion addressing this issue (lines 673-677): *“The observed mild rescue however indicates differences between the two proteins regarding their specificities for target RNAs or cofactors. Also the dsRBD1-5 of the house fly (Musca domestica) Stau failed to rescue all aspects of oskar mRNA localization (Micklem et al, 2000). It will be interesting to see the basis of these functional differences in future experiments.”*

Minor comments:

1. Figure S1: Figure legend for the schematic is missing and I would move this to Fig 1 since I think this information is helpful for interpreting the experiments throughout the paper.

We are grateful for this comment and added the figure legend accordingly. The schematic representations of the RNAs are all present in small size in the main Figure 1 and in a much larger version in Supplementary Figure 1. We tried to transfer the larger schematic drawings into the main figure and realized that this spoils the entire format of this main figure. We therefore decided to refrain from adopting this suggestion.

2. Figure S3: Electron density map (2Fo-Fc?) and sigma contour should be stated

We added the requested information to the legend of Supplementary Figure 3.

3. Figure S8: a mapping of the mutations onto a dsRBD structure would be helpful

We performed homology modelling using the PHYRE2 algorithm for dsRBD1 and for dsRBD2. While for the first one, the model had a reasonable quality, for the latter it did not. We therefore decided to plot the mutations only on the homology model of dsRBD1. We hope Reviewer 1 agrees with our reasoning. Figure S8 (now Supplementary Figure 9) has been changed accordingly.

4. Figure S9: Referenced out of order in the text

We changed the order of the supplementary figures and corrected this mistake in the text. The former Figure S9 is now Supplementary Figure 7.

5. Figure S10-14: I find the star marking the position of the mutation in the dsRBD domain to be a little confusing because a single star is used to denote multiple different mutations in the schematic. I would try to clarify this with labeling of the mutation or the use of different symbols. Also labeling in panels and figure legend are not consistent in Fig S10, S11,

We tried to implement the suggested changes but found that different labels for the mutations in the Supplementary Figures 10 onwards render the figures rather confusing. We therefore refrained from adopting these changes and hope that Reviewer 1 agrees with our decision.

We corrected the figure legends to match the panels.

6. line 539 should refer to figure 1F and not 1E

We appreciate this comment and have corrected this mistake.

Reviewer #2 (Remarks to the Author):

This article describes biochemical and functional studies of the murine protein Staufen2, in particular with respect to the mRNA binding capacity of its two domains

dsRBDs 1 and 2. These domains had been reported previously not to bind RNA, while the RNA-binding capacity of Staufen had been attributed exclusively to dsRBDs 3 and 4. Using EMSA, SPR and NMR, the authors demonstrate that dsRBDs 1-2 also contribute to RNA binding. This conclusion is fully supported by the experiments and unquestionable. However, I have many problems with the quantitative interpretation of nearly all experiments and cannot recommend publication of this article in the present form. In details:

1. The EMSAs of Figure 1 show that SRS2, and SRS2+5 are shifted in two distinct bands depending on the protein concentration by both dsRBDs 1-2 and dsRBDs 3-4. This “double shift” cannot be recapitulated by the subsequent binding of the two dsRBDs present in each construct, as an intermolecular binding of the second domain of the tandem dsRBDs, following the binding of the first domain, should not depend on the concentration of the protein. The authors do not comment on this.

We would like to point out that the double-band is indeed not only observed in Figure 1 but consistently also in all subsequent EMSAs with tandem dsRBDs (see Supplementary Figure 5). This could either mean that a second molecule joins the complex or that a conformational change occurs at higher concentrations. Since we cannot distinguish between these two possibilities and Reviewer 2 does not explicitly asks for including such a comment in the manuscript, we took the liberty not to elude on this in the manuscript and thereby avoid what we consider a potential overinterpretation.

2. On line 459, the line broadening of the NMR signals is interpreted in terms of sliding of the protein on the RNA. This is not in agreement with the localization of the line broadening to the last four base pairs (line 436-437). If the protein slides along the long stem of SRS2+5, thus contributing an entropic factor to the binding energy, and if this sliding translates into imino line broadening, why should 4 base pairs at the terminus be preferentially affected?

Thank you for this careful analysis. Indeed, we believe that by comparing the binding to these two RNAs additional insight is gained. Please note, that the main interaction site was mapped for the shorter SRS2 RNA based on most significant line-broadening observed for the terminal base pairs. The fact that most signals remain observable upon binding to SRS2 RNA suggests that there is no or little sliding involved, presumably due to the shorter length of the stem, indicating that a

12 bp stem-loop is too short to allow for sliding. In contrast the 17 bp stem in SRS2+5 RNA shows severe line-broadening beyond detection for all imino signals, which suggests exchange-mediated line-broadening due to sliding. We added the following statement in the manuscript to clarify this point (lines 210-230): *“With the shorter SRS2 stem-loop RNA, all imino signals are observable at equimolar RNA:protein ratio. Line-broadening for the imino signals in the base pairs at the bottom of the stem suggests this as a main interaction region. In contrast, the 17 bp stem of SRS2+5 allows for significant sliding as reflected by the severe line-broadening observed for all imino signals in the base pairs of the stem upon protein binding.”*

3. A 7 base pair stem is considered to be the minimum sequence that binds the tandem dsRBDs. However, the SPR experiments were performed on a much longer sequence, which then potentially contains multiple binding sites. I would believe the conclusions from the following SPR experiments much more if they had been performed on this minimal construct rather than on the longer SRS* or SRS2+5 constructs.

The choice of dsRNA for SPR experiments was motivated by the physiological relevance of the RNA stem-loops. Literature suggests that physiologic RNA targets contain stem loops of at least 12 bp length (SRS RNAs: Laver et. al, 2013 and Heraud-Farlow et. Al, 2013). This is in full agreement with the statement of Reviewer 1, who pointed out the importance of using stem loops that are not too short (see Comment 2 of Reviewer 1).

Furthermore, we wanted to be able to detect simultaneous binding by up to four dsRBDs, which we do not expect to be possible with such a short 7 bp stem-loop. Such a short RNA does not seem suitable to detect the differences between binding by a tandem-dsRBD vs. four dsRBDs and thus for detection of the stable complex. Thus, the shortening of the stem loop answered the question of the minimal length requirement for binding of tandem dsRBDs but is not suited to disentangle the respective contributions of individual dsRBDs in larger protein context.

4. The binding curves of dsRBDs 1-2 and dsRBDs 3-4 to the SRS* and SRS2+5 RNAs show a two-site binding behavior. Later (line 710-713), it is suggested that these two events correspond to RNA binding of the first and the second dsRBD in the

tandem construct. This is surprising, as binding of tandem domains to a tandem substrate (the RNA) should not depend on protein concentration. The local concentration of the second binding event depends only on the saturation of the first binding event. These two binding events may represent the binding of two copies of dsRBDs 1-2 or dsRBDs3-4 to the RNA. The meaning of the two binding events is not discussed in the article and the suggestion of line 710-713 is in my opinion wrong.

The statement in our original manuscript (lines 710-713) is the following: “*Our comparison of RNA binding affinities of single domains and tandem domains bearing mutations suggest that the first binding event with moderate affinity is achieved by the second dsRBD in each tandem domain, namely dsRBD2 and dsRBD4.*” As can be seen, we do not state that the interpretation of the two-site binding is the subsequent binding by dsRBD2 and then dsRBD1. We fully agree that such a statement would have been an over interpretation.

The statement that our data suggest initial binding by dsRBD2 and dsRBD4 is based on the much higher affinities of dsRBD2 and dsRBD4 over dsRBD1 and dsRBD3, on affinities of the tandem domains and on mutational studies of domains in the greater multidomain context. We still consider this interpretation sound enough to propose this model.

The same is true for the interpretation of the Hill coefficient of 1.7 as a cooperative behavior of the four domains belonging to the same protein. What excludes some “cooperativity” between two protein copies? In addition, interpreting the Hill coefficient as a sign of cooperativity is an oversimplification of more complicated, case-specific models. The authors should apply these models to interpret the binding curve, considering that the local concentration of the second, third and fourth dsRBD is influenced by binding of the first one. This might be very complicated; on the other hand a wrong over simplification is not of much help.

We do agree that RNA binding of the four dsRBDs is a complicated issue. With regards to the observed Hill coefficient, it is beyond doubt in the field that this can only be interpreted as cooperativity. This *per se*, in our eyes, is no oversimplification. The question behind it is if a more complicated interpretation is rather more accurate or an over interpretation. Here we seem to disagree.

We also agree that we cannot formally exclude cooperativity via protein-protein interaction. Since dimerization has been previously reported for Stau1 (Gleghorn et al. *Nat Struct Mol Biol* 2013), we now include static light scattering experiments

showing that full-length Stau2 does not dimerize (Supplementary Figure 6). We therefore consider a scenario of two tandem domains binding cooperatively to RNA more obvious and likely than an intermolecular interaction. In order to account for the concerns of Reviewer 2, we rephrased this interpretation by stating that protein-protein interactions are an option, although unlikely. We also went through the entire manuscript and toned down our statements on cooperativity wherever ambiguous.

New statement (lines 284-290): *“Whether this cooperativity arises from interactions of individual dsRBDs within one protein or from protein-protein interaction between different molecules cannot be determined. The previously reported dimerization of Stau1⁴⁸, suggests that Stau2 might also form oligomers. We did, however, not detect oligomerization of Stau2 by SEC-SLS (Supplementary Figure 6) and thus consider cooperativity by intermolecular interactions unlikely.”*

We are aware of a recent report by Andres Ramos and colleagues (Nicastro et al., 2017), in which the suggested mathematical modelling has been applied to understand the RNA binding of KH domains 3 and 4 of the ZBP1 protein. In contrast, our RNA binding by Stau2 involves four dsRBDs. We do not see how this could be reasonably modelled with the data at hand. We are therefore convinced that mathematical modelling of all four binding events would constitute an over interpretation. We understand that Reviewer 2's request aimed at strengthening our study, but decide to refrain from what we consider an over interpretation of our data.

Regardless of the source of cooperativity, the fact that only a construct with both tandem domains (dsRBD1-4) shows stable RNA binding is undisputed also by Reviewer 2. This is the most important message from these experiments.

5. In the NMR analysis of isolated dsRBD1 and dsRBD2 domain it is noticed that dsRBD1 is present in several different conformations. From the text it remains unclear whether this conformational multiplicity is present also in the tandem dsRBDs 1-2 construct. Inspection of Fig. S6 and comparison with Figure 2 shows that this is indeed the case.

We are grateful for this comment. The comparison of NMR spectra of individual and tandem domains shows that there are essentially no relevant differences. The fact that for some amides in dsRBD1 two sets of signals are observed is now

mentioned in the legend of Figure 3 (lines 1243-1246): “*Note, that there are two sets of signals observed for dsRBD1, where only one set is affected by RNA binding. The second set of signals may reflect the presence of an alternate conformation of a region of dsRBD1.*” We also addressed this issue in the main text (lines: 311-314): “*This indicates that in the context of the tandem domains the structures of the individual dsRBDs 1 and 2 are not altered and do not significantly interact with each other.*” However, as we have not been able to obtain complete backbone assignment for the protein, the origin of the second set of signals is not clear. It may reflect peptidyl-prolyl cis/trans isomerization. We therefore would not like to further speculate on this.

Nevertheless, on line 539 it is written “Interestingly, for dsRBD1 less line-broadening is observed for imino signals at a protein:RNA ratio of 1:0.5. This indicates that the two dsRBDs bind the RNA with different dynamics and possibly reduced affinity...” I am surprised of such an observation. The authors seem to forget that, if the protein is present in different conformations, only one of these conformations is competent for binding (as shown also in Fig. 2A), thereby reducing the effective protein concentration!!!! The observation that differential line broadening at the same RNA:protein ratio indicates different dynamics is, in this context, simply wrong.

We agree with the general argument of Reviewer 2 that a reduced concentration of active protein conformation reduces the apparent RNA binding affinity, but we respectfully disagree with the concern raised for a number of reasons. 1) It appears that only a small set of amide groups shows two sets of signals, thus suggesting that this is not an overall effect of two protein species, but only a local effect of perhaps an alternate local conformation. 2) Moreover, the 1D imino NMR spectra of the RNA in the presence of dsRBD1 and dsRBD2 also show very different line-broadening, consistent with a reduced binding affinity by dsRBD1. This is true even if the “active” dsRBD1 protein would be only 50% of the total concentration, as the line-broadening observed at 1:1 molar ratio dsRBD1:RNA is still much less than observed at 0.5:1 molar ratio of dsRBD2:RNA. 3) Finally, Figure 2a shows Biacore experiments that also indicate a reduced binding affinity for dsRBD1.

We observed strong line broadening in a concentration-dependent manner only for dsRBD2. For dsRBD1, imino signals remain sharp with increasing protein concentrations and no effect can be observed. In order to clarify this issue, we changed this sentence to (lines 322-325) “*Interestingly, for dsRBD1 less line-broadening is observed for imino signals (Figure 3B). This indicates that the two*

dsRBDs bind RNA with different binding kinetics and suggestive of reduced binding affinity of dsRBD1 compared to dsRBD2.” In our eyes, this indicates different modes of binding.

6. On the same page, the RNA binding activity of dsRBD1 and dsRBD2 are compared on line 545-573. This paragraph ignores that the multiple conformations observed for dsRBD1 may be an artifact of the truncated construct. The presence of these multiple conformations diminishes the binding-competent concentration of protein. Thus, it remains unclear whether the very weak binding activity of dsRBD1 is a property of dsRBD1 in the context of the full protein.

Concerning the validity of our conclusion that dsRBD1 has intrinsically reduced RNA binding affinity please refer to our comment to the previous point. Given the similarity of NMR chemical shifts for signals of individual domains (including dsRBD1) in isolation vs the tandem dsRBD protein, we conclude that the general features of RNA binding seen for the individual domains are also seen in the tandem domain protein.

7. Discussion of dsRBD1 mutations, line 591 to 613. Which should be the mechanism by which the length of the RNA compensates a mutation in the canonical RNA binding site of a dsRBD?

We are grateful for this comment, we have removed this statement altogether.

8. The dsRBD2 mutation H169A has a more dramatic effect on RNA binding than F157A when tested in the dsRBDs 1-2 construct. However, this mutation becomes less deleterious than F157A when seen in the context of the dsRBDs 1-4 and together with the F40A mutation. Why?

While F40A F157 in dsRBD1-4 clearly compromises the on-rate of the RNA binding, the F40A H169A mutations alter the kinetics of the interaction more dramatically (Figure 5). For instance, RNA binding by dsRBD1-4 F40A F157A did not show the same transient nature of the dsRBD3-4 interaction. In contrast, dsRBD1-4 F40A H169A this is clearly the case. This indicates that the RNA binding activity of dsRBD1-2 may not completely be abolished in the F40A F157A mutant. The mutation is thus weaker also in the dsRBD1-4 context. In order to fully explain the molecular basis of this observation, co-structures of each mutant dsRBD1-4 in complex with RNA would be required. This would be clearly beyond

the scope of this manuscript. Understanding these features would also not make a significant contribution to our study.

9. The ability of mStau2 to complement DmStau is only moderate. In light of this, I wonder how significant the functional assays of Fig. 6 are. Clearly, this moderate complementation ability may indicate that the mode of recognition of mStau2 versus the DmRNA might differ from the native one. In this context, what are the experiments with the mutant mStau2 telling us?

Although the rescue by mStau2 of *oskar* mRNA being moderate, the data are statistically significant. This tells us, despite likely differences in specificity or mRNP biogenesis, that the physiological Staufen-target mRNA is also bound by its mouse paralogue and that there is likely functional conservation. Of note, even the housefly homologue of DmStaufen fails to rescue all aspects of *oskar* mRNA localization (no posterior accumulation during stage 9, see Micklem et al., EMBO J., 2000). More importantly, however, the partial rescue by wild-type mStau2 is used as a reference for mutational studies, assessing the importance of RNA binding by dsRBD1-2 *in vivo*. Thus, it serves as cross-validation and final confirmation that our *in vitro* findings are relevant in a physiologic context.

We indeed plan to follow up on this observation in our future work. We also added a statement in the Discussion addressing the issue of the mild rescue (lines 673-677): “*The observed mild rescue however indicates differences between the two proteins regarding their specificities for target RNAs or cofactors. Also the dsRBD1-5 of the house fly Stau (Musca domestica) failed to rescue all aspects of oskar mRNA localization (Micklem et al, 2000). It will be interesting to see the basis of these functional differences in future experiments.*”

We also added a more detailed statement on the specificity (lines 654-660): “*While we cannot exclude specificity of dsRBDs for certain sequence motifs, we found no experimental evidence for such an assumption. Our data rather suggest that the Stau2 protein recognizes its RNA target in a structure- and not sequence-dependent manner. Likely scenarios for the specificity reported in vivo include the recognition of combinations of secondary structure elements or a contribution of cofactors.*” We hope that these additions resolve this issue.

REVIEWERS' COMMENTS:

Reviewer #1 (Remarks to the Author):

The authors have made all appropriate modifications to their manuscript and I now support its publication.

Reviewer #3 (Remarks to the Author):

As noted by Reviewers 1 and 2, this manuscript establishes that all 4 dsRBDs of Staufen, and not just dsRBD3 and dsRBD4, contribute to RNA binding in vitro and provide evidence that they also contribute to function in vivo. Thus, this work provides a significant contribution to understanding Staufen function and will be very useful for the field. Both reviewers asked for extensive clarifications, and in particular reviewer 2 was concerned about the quantitative interpretation of the experiments. For the most part, the authors have made reasonable responses to the criticisms. However, in my opinion, much of the problem that the previous reviewers had stems from the lack of rigour in how results are described (imprecise wording and in some cases incorrect English language usage), and figures that are not always clear. These problems remain in the revised manuscript. I suggest that the manuscript be carefully edited for clarity and accuracy in language so that this paper has the impact that the data warrant; some suggestions follow:

1. Some examples of problems with the figures:

Figure 1C (described on page 6) compares the imino spectra of free SRS2 and a 1:1 complex with Stau2 dsRBD 1-2. First, binding of the protein would be expected to broaden the lines simply from the increase in MW. There can be additional line broadening from chemical exchange (although one doesn't expect too much change in the iminos for a dsRBD since the contacts are mostly to the backbone). The authors say that some peaks are more broadened than others, but this is simply impossible to see from the way the spectra are plotted, we have to take their word for it. The spectra could be plotted over a narrower chemical shift range (i.e. to 13.8 ppm) and twice as high and also the linewidths could be reported.

Also, why are the chemical shift scales labeled differently in Fig 1c and Fig 1f (right)? What is the top black line of noise?

Figure 1f and 3 a,c: The superimposed HSQC's are plotted with the contours all blurred into solid peaks, such that it is impossible to see the changes wherever peaks stack on each other.

SI figs: It is almost impossible to read the nucleotide labels in SI Fig 1A, SI Fig 5, etc. Also, SI Fig 4, why are there giant numbers on the side of the sequence?

These suggestions may seem picky, but it gives the reader a sense of sloppiness that may extend to the data analysis and interpretation.

2. On page 7, the SRS2+5 is described as 17 bp, by my count it is 18 plus possibly a loop bp.

3. Also on page 7: The authors conclude (line 194-195) that "a stem of 7 bp appears to be the minimal length required for recognition by mStau2 tandem domains." This doesn't make sense to me. The dsRBD binds successive minor, major, minor grooves on one face of an RNA helix, so the minimum binding site has to be ~15 bp. Please clarify.

4. Abstract: How can RNA binding by tandem domains be transient, while all four dsRBDs recognize their target RNAs with high stability, as stated in the abstract. I strongly suggest to rewrite this sentence. The dsRBD1-2 mutant mStau2 referenced in the abstract does not convey much information; why not say mStau2 dsRBD1-2 with mutations that abolish binding....??

5. Regarding the response to Reviewer 2, point 1: I agree with the reviewer that a comment on the supershift needs to be made (as least to acknowledge that it is there), especially as it is consistently seen in other EMSAs.

6. Regarding the response to Reviewer 2, point 2: It might be worth noting that since you are using a hairpin, the dsRBD probably doesn't slide off the hairpin end of the RNA but rather is stopped there, which can explain the differential line broadening.

7. Regarding the response to Reviewer 2, point 5: If the doubled peaks are due to something like cis-trans isomerization, shouldn't the equilibrium be shifted to the bound conformation as protein is added?

8. I found the discussion of the putative mStau2 binding sites somewhat confusing. Do the authors think dsRBD1-2 binds one stem-loop and dsRBD 3-4 binds a second stem loop? Or do both bind to one long stem-loop?

In this regard, it might improve the discussion to compare how dsRBDs bind to dsRNA stem-loops in other examples in the literature.

REVIEWERS' COMMENTS:

Reviewer #1 (Remarks to the Author):

The authors have made all appropriate modifications to their manuscript and I now support its publication.

We are grateful that reviewer #1 supports publication of our revised manuscript in its present form.

Reviewer #3 (Remarks to the Author):

As noted by Reviewers 1 and 2, this manuscript establishes that all 4 dsRBDs of Staufen, and not just dsRBD3 and dsRBD4, contribute to RNA binding in vitro and provide evidence that they also contribute to function in vivo. Thus, this work provides a significant contribution to understanding Staufen function and will be very useful for the field. Both reviewers asked for extensive clarifications, and in particular reviewer 2 was concerned about the quantitative interpretation of the experiments. For the most part, the authors have made reasonable responses to the criticisms. However, in my opinion, much of the problem that the previous reviewers had stems from the lack of rigour in how results are described (imprecise wording and in some cases incorrect English language usage), and figures that are not always clear. These problems remain in the revised manuscript. I suggest that the manuscript be carefully edited for clarity and accuracy in language so that this paper has the impact that the data warrant; some suggestions follow:

We thank reviewer 3 for his balanced view and suggestions. We reassessed all figures and carefully checked the manuscript for inconsistencies, imprecise wording and unclarities in the figures and text. We hope that with the revised manuscript we could address the remaining concerns and satisfy the requests of Reviewer #3.

1. Some examples of problems with the figures: Figure 1C (described on page 6) compares the imino spectra of free SRS2 and a 1:1 complex with Stau2 dsRBD 1-2. First, binding of the protein would be expected to broaden the lines simply from the increase in MW. There can be additional line broadening from chemical exchange (although one doesn't expect too much change in the iminos for a dsRBD since the contacts are mostly to the backbone). The authors say that some peaks are more broadened than others, but this is simply impossible to see from the way the spectra

are plotted, we have to take their word for it. The spectra could be plotted over a narrower chemical shift range (i.e. to 13.8 ppm) and twice as high and also the linewidths could be reported. Also, why are the chemical shift scales labeled differently in Fig 1c and Fig 1f (right)? What is the top black line of noise?

We agree that an overall line width increase will be caused by an increase in molecular weight. Additional contributions to line broadening from chemical exchange however are also plausible since binding of a dsRBD to the dsRNA backbone and potential dynamics of the complex will affect the chemical environment of the imino protons. We and others have frequently observed this with other dsRBD proteins (i.e. Tants et al NAR 2017). As indicated by reviewer #3, Fig. 1c reveals that the overall line width of the complex (red) is larger than for the free RNA. However, there is an additional line-broadening observed for U29, U27 and G5/G11. We thank the reviewer for his/her suggestions to improve the presentation of the NMR data and have changed the following:

- We have replotted the spectra as suggested by the reviewer to make this differential line-broadening better visible by showing a more zoomed-in view. We also explain in the Methods section on page 23 how Fig. 1c was plotted, i.e. the bound spectrum was scaled so that imino signals, which do not show additional exchange-mediated line-broadening in the complex, have the same peak height as in the free spectrum (still with larger line-broadening). An integration of the signal intensities from the 1D spectra is shown here:

Ratio of 1D RNA imino signal intensities of a 1:1 Stau2 dsRBD1-2-RNA complex to free SRS2 stem-loop RNA. *Intensities in the two spectra were normalized against the signal of the U25 imino proton. ¹in wobble base pairs, only the uracil residues were evaluated due to overlap in the guanine resonances ²line broadening leads to signal overlap in the complex spectrum

- The spectral axis are now consistently labeled in all figures as “¹H chemical shift [ppm]” etc.
- As indicated in Fig. 1f (right) the black line is the free protein before addition of RNA plotted as reference.

Figure 1f and 3 a,c: The superimposed HSQC's are plotted with the contours all blurred into solid peaks, such that it is impossible to see the changes wherever peaks stack on each other.

We thank reviewer #3 for pointing this out. This likely reflects issues in the pdf conversion. We have replotted the contours, updated both figures and ensured that contours are better visible in the PDF conversion.

Note that the purpose of Fig 1f is indeed to show the severe line-broadening associated with binding, which essentially renders many of the NMR signals in the green spectrum not detectable. For Fig. 3a,c we have replotted and checked that contours are indeed visible in the PDF version. In Figure 3c, we also added an inset with a magnification of an area to allow for a better assessment of the data. In any case, all figures of this revised manuscript have been uploaded as EPS file with vector graphics, allowing for full scaling of the results.

SI figs: It is almost impossible to read the nucleotide labels in SI Fig 1A, SI Fig 5, etc. Also, SI Fig 4, why are there giant numbers on the side of the sequence? These suggestions may seem picky, but it gives the reader a sense of sloppiness that may extend to the data analysis and interpretation.

We are grateful for this notion. As suggested, we re-evaluated all Supplementary Figures and changed the following items:

- Suppl. Fig. 1a: subfigures are now larger, including larger nucleotide labels
- Suppl. Fig. 4: large numbering of the nucleotides have been reduced in size
- Suppl. Fig. 5: same as in Suppl. Fig. 1a.

Furthermore, we found issues in the following figures and corrected them as well:

- Suppl. Fig. 3: subfigure size was changed and stereo image of electron density was added.
- Suppl. Fig. 7: contour levels were optimized and higher resolution image taken.
- Suppl. Fig. 14: box around the legends and double-labeling of wt concentrations have been removed.

- Supp. Fig. 18: we noted that during PDF conversion some of the concentration labels of the EMSAs had disappeared. This has been corrected.

2. On page 7, the SRS2+5 is described as 17 bp, by my count it is 18 plus possibly a loop bp.

Indeed, the description of the number of base pairs in the text deviated by one base from the figure. This was not only the case for the SRS2+5 but also for shortened versions of this RNAs (see Supplementary Figure 5b-d). We corrected it throughout the text and figure, and apologize for this inconsistency.

3. Also on page 7: The authors conclude (line 194-195) that “a stem of 7 bp appears to be the minimal length required for recognition by mStau2 tandem domains.” This doesn’t make sense to me. The dsRBD binds successive minor, major, minor grooves on one face of an RNA helix, so the minimum binding site has to be ~15 bp. Please clarify.

Available structures indeed suggest a minor-major-minor groove association of dsRBDs. However, our EMSAs clearly show that also shorter stem loops can be recognized by dsRBDs 1-2 or 3-4. Thus our findings cannot be easily explained by existing co-structures. We now added the following sentences to the discussion to speculate about the reason for this observation:

“Of note, available structures of dsRBDs show binding to longer RNA stem-loops of about 19 bp length^{48, 49}. It is therefore well possible that our observed binding to a minimal stem-loop RNA only reflects a partial recognition and that for a full binding a longer stem is required. This interpretation is consistent with our general observation that longer RNAs are bound stronger than shorter ones.”

4. Abstract: How can RNA binding by tandem domains be transient, while all four dsRBDs recognize their target RNAs with high stability, as stated in the abstract. I strongly suggest to rewrite this sentence. The dsRBD1-2 mutant mStau2 referenced in the abstract does not convey much information; why not say mStau2 dsRBD1-2 with mutations that abolish binding....??

We agree that this statement was too cryptic. This sentence has now been changed into the following statement: “In contrast, a rescue with mStau2 bearing

RNA-binding mutations in dsRBD1-2 fails,...". In order to stay within the 150 words limit, we also had to slightly change the first sentence of the Abstract.

5. Regarding the response to Reviewer 2, point 1: I agree with the reviewer that a comment on the supershift needs to be made (as least to acknowledge that it is there), especially as it is consistently seen in other EMSAs.

As requested by reviewers #2 and #3, we added a statement about the supershift on page 5: "Upon increased protein concentrations a supershift was observed, indicating either binding of additional dsRBDs to the RNA or an oligomerization of the protein itself."

6. Regarding the response to Reviewer 2, point 2: It might be worth noting that since you are using a hairpin, the dsRBD probably doesn't slide off the hairpin end of the RNA but rather is stopped there, which can explain the differential line broadening.

Thank you for this comment. Indeed, the fact that the proteins binds to a hairpin not a duplex RNA is also expected to introduce some asymmetry in the binding and sliding. We have added the following statement in this respect in the text (lines 187-190):

"Because the protein may not slide off the hairpin end but rather gets stopped there, protein binding to a hairpin RNA is expected to introduce some asymmetry to binding and thus differential line broadening."

7. Regarding the response to Reviewer 2, point 5: If the doubled peaks are due to something like cis-trans isomerization, shouldn't the equilibrium be shifted to the bound conformation as protein is added?

Thank you for this comment. It will depend to what extent the residues with doubled peaks are in the RNA binding interface and if only one of the conformations is competent in RNA binding. As the origin of the doubled peaks is not fully understood and likely linked to the fact that it concerns a truncated single domain we would rather not further speculate on this.

8. I found the discussion of the putative mStau2 binding sites somewhat confusing. Do the authors think dsRBD1-2 binds one stem-loop and dsRBD 3-4 binds a second stem loop? Or do both bind to one long stem-loop? In this regard, it might improve

the discussion to compare how dsRBDs bind to dsRNA stem-loops in other examples in the literature.

We thank reviewer #3 for this comment, as it precisely reflects one of the key questions we try to resolve in our lab in the near future. We added the following statement and hope that this addresses the above raised concern: “Future work will have to answer whether a defined spatial arrangement of two stem-loops is recognized by each tandem dsRBD or if all four domains act as a molecular ruler for a single stem-loop of defined length.”

Since to our knowledge no structure is available on a protein with four dsRBDs bound to a structured RNA, we decided to refrain from referring to literature on this issue.